# Potent human neutralizing antibodies against Nipah virus derived from two ancestral antibody heavy chains

Li Chen [1,2,10], Mengmeng Sun [3,10], Huajun Zhang [1,10], Xinghai Zhang[1,10], Yanfeng Yao [1,10], Ming Li[3], Kangyin Li[1,2], Pengfei Fan [4], Haiwei Zhang[1], Ye Qin[1,2], Zhe Zhang[1], Entao Li[3,5,6], Zhen Chen[1], Wuxiang Guan[1], Shanshan Li [3], Changming Yu [4] ✉, Kaiming Zhang [3,6,7,8] ✉, Rui Gong [1,2,9] ✉ & Sandra Chiu [3,5,6] ✉

Nipah virus (NiV) is a World Health Organization priority pathogen and there are currently no approved drugs for clinical immunotherapy. Through the use of a naïve human phage-displayed Fab library, two neutralizing antibodies (NiV41 and NiV42) targeting the NiV receptor binding protein (RBP) were identified. Following affinity maturation, antibodies derived from NiV41 display cross-reactivity against both NiV and Hendra virus (HeV), whereas the antibody based on NiV42 is only specific to NiV. Results of immunogenetic analysis reveal a correlation between the maturation of antibodies and their antiviral activity. In vivo testing of NiV41 and its mature form (41-6) show protective efficacy against a lethal NiV challenge in hamsters. Furthermore, a 2.88 Å Cryo-EM structure of the tetrameric RBP and antibody complex demonstrates that 41-6 blocks the receptor binding interface. These findings can be beneficial for the development of antiviral drugs and the design of vaccines with broad spectrum against henipaviruses.

Nipah virus (NiV), a zoonotic, single-stranded RNA virus in the genus *Henipavirus*, family *Paramyxovirus*, has been shown to spillover from its natural host, *Pteropus* bats, to other mammals, including humans[1–3]. NiV can be transmitted to humans through reservoir animals, contaminated food, or other humans[4]. The infections can range from asymptomatic to acute respiratory infection and even fatal encephalitis. Since the initial outbreak in 1999 in Malaysia[5], NiV spillovers have been recurrently reported in certain parts of Asia, mainly Bangladesh[4,6]

and India[7]. It separates into two distinct lineages/genotypes: Nipah-Malaysia (NiV_M) and Nipah-Bangladesh (NiV_B)[8]. Although the mortality rate from NiV infection can range from 40 to 90%[9], there is still no approved treatment for NiV. In 2017, the World Health Organization (WHO) designated NiV as a blueprint priority disease that poses a significant public health risk. However, there is no approved vaccine or drug on the market yet[1]. Hendra virus (HeV), a fatally zoonotic member of the genus *Henipavirus*, was first identified in Australia in 1994. It has

[1]CAS Key Laboratory of Special Pathogens and Biosafety, Wuhan Institute of Virology, Center for Biosafety Mega-Science, Chinese Academy of Sciences, Wuhan, Hubei, China. [2]University of Chinese Academy of Sciences, Beijing, China. [3]Division of Life Sciences and Medicine, University of Science and Technology of China, Hefei, Anhui, China. [4]Laboratory of Advanced Biotechnology, Beijing Institute of Biotechnology, Beijing, China. [5]Department of Laboratory Medicine, The First Affiliated Hospital of USTC, Division of Life Sciences and Medicine, University of Science and Technology of China, Hefei, Anhui, China. [6]Key Laboratory of Anhui Province for Emerging and Reemerging Infectious Diseases, Hefei, China. [7]Center for Advanced Interdisciplinary Science and Biomedicine of IHM, MOE Key Laboratory for Cellular Dynamics, Division of Life Sciences and Medicine, University of Science and Technology of China, Hefei, Anhui, China. [8]Department of Urology, The First Affiliated Hospital of USTC, Division of Life Sciences and Medicine, University of Science and Technology of China, Hefei, Anhui, China. [9]Hubei Jiangxia Laboratory, Wuhan, Hubei, China. [10]These authors contributed equally: Li Chen, Mengmeng Sun, Huajun Zhang, Xinghai Zhang, Yanfeng Yao. ✉e-mail: yuchangming@126.com; kmzhang@ustc.edu.cn; gongr@wh.iov.cn; qiux@ustc.edu.cn

caused sporadic outbreaks in Australia, leading to seven human infections, including four cases of death due to fatal encephalitis, giving a mortality rate of 57%[10]. It has been determined that human infections are linked to close contact with sick horses infected by HeV.

NiV contains two viral surface proteins, receptor binding protein (RBP, also known as G protein) and fusion glycoprotein (F protein)[11], which mediate viral entry into host cells and its spread from cell to cell within infected hosts. Upon binding to the host cell receptor, the transmembrane protein tyrosine kinases ephrinB2 or ephrinB3[12,13], RBPs undergo conformational changes, triggering F protein to shift from a metastable prefusion form to a highly stable postfusion form[14–16]. This process involves an extensive structural rearrangement resulting in virus–cell or cell–cell membrane fusion.

NiV-RBP is a type II membrane glycoprotein featuring a brief N-terminus cytoplasmic tail, a single transmembrane domain, and a sizable ectodomain. This ectodomain is composed of a stalk region and a C-terminus receptor binding globular head domain, linked by a pliable neck to the helical stalk[17]. The RBP is arranged as a homo-tetramer within each protomer linked by disulfide bonds in the neck and stalk region[18,19]. The structure of the stalk region includes a parallel four-helix bundle. Biochemical studies have indicated that the stalk region is important for oligomeric stability, F protein interaction, and specificity[20]. The globular head domain structure of RBP is related to those of the other paramyxovirus attachment proteins with a six-β-sheet propeller structure[17,21,22]. Monoclonal antibodies, as a highly effective immunotherapeutic treatment against viral infections, are now widely used for pre- or postexposure protection against infectious diseases caused by highly virulent and lethal viruses, such as Ebola virus (EBOV)[23,24] and severe acute respiratory syndrome coronavirus 2 (SARS-CoV-2)[25,26]. Previous studies have demonstrated that the effective neutralizing antibodies currently developed against NiV mainly focus on RBPs[18], such as m102.4[22,27,28], a phage display-derived antibody that is being clinically evaluated, and HENV26[29], which was isolated from an individual who received the equine HeV-RBP subunit vaccine (Equivac) against HeV.

Here, we identified two promising candidate human monoclonal antibodies (41-6 and 42-27) with potent neutralizing activity through a process of antibody phage display library screening and affinity maturation. An alanine scanning mutagenesis assay was performed, and the results showed that two antibodies recognized separate epitopes. Furthermore, we found that the maturation pattern of antibody genes was connected to neutralizing activity and antiviral breadth. A 2.88 Å cryo-electron microscopy (cryo-EM) structure of the dimeric head region of the full tetramer NiV-RBP complexed with 41-6 Fab fragments revealed that the Fab functions by obstructing the receptor's binding. This antibody also exhibits protective efficacies in hamster models of chronic and acute diseases, demonstrating its potential as both prophylactic and therapeutic agents. In addition, the epitope information disclosed here could also be significant for the development of vaccines and drugs against henipaviral infections.

## Results

### Identification of two neutralizing antibodies from the phage display library

By panning the human naïve Fab phage display library against NiV-RBP-Fc, we identified two clones, FabNiV41 and FabNiV42, which exhibited robust binding capacities to antigen NiV-RBP-Fc without any reactivity to the control protein, bovine serum albumin (BSA). The two clones bound to NiV-RBP with $EC_{50}$ values of 0.2 and 0.1 µg/ml, respectively (Fig. 1a). Following a biolayer interferometry (BLI) binding experiment (Fig. 1b), it was ascertained that FabNiV41 is capable of binding with RBPs from both NiV and HeV, although the binding to NiV is stronger with a slower dissociation rate. Meanwhile, FabNiV42 has a significantly stronger binding capacity with the NiV-RBP, reaching the sub-nanomolar level, but its HeV-RBP binding ability is dramatically

weaker. The results from the flow cytometry experiment demonstrated that both antibodies could prevent the binding of soluble antigen protein (RBP) to the surface receptor in the sensitive cell membrane (Fig. 1c), suggesting that the two Fab clones had potential neutralization ability. We converted the clones into IgGs as NiV41 and NiV42, respectively, and examined their capacities. NiV41 demonstrated a comparable binding to the NiV-RBP with NiV42, but a much stronger binding to the HeV-RBP than NiV42 (Fig. 1d). We used the pseudovirus system to evaluate the neutralization capacity of the two antibodies (Fig. 1e). Both exhibited strong neutralization of NiV, with $IC_{50}$ values of 0.07 and 0.02 µg/ml, respectively. In addition, NiV41 and NiV42 also displayed neutralization against HeV with comparable activities. The authentic virus neutralization assay also indicated that NiV41 had cross-neutralization capability against NiV and HeV ($NiV_M$ $IC_{50} = 0.37$ µg/ml, $NiV_B$ $IC_{50} = 1.09$ µg/ml, HeV $IC_{50} = 0.77$ µg/ml). In the case of NiV42 neutralized $NiV_M$, $NiV_B$, and HeV, the $IC_{50}$ values were 0.19, 0.67, and 0.39 µg/ml, respectively (Fig. 1f). NiV41 was also selected to preliminarily evaluate its protective efficacy against $NiV_B$ in chronic-infection hamster models[30]. Our results indicate that administration at a dose of 3 mg/kg, six hours after virus exposure, provided significant protection to the infected animals (Fig. 1g).

### Isolation and characterization of a panel of potent neutralizing antibodies through affinity maturation

To bolster the neutralizing potency of these two antibodies, affinity maturation was implemented by utilizing light chain shuffling[31]. After four rounds of panning against henipaviral RBPs, 22 clones that exhibited strong binding capabilities ($EC_{50} < 0.5$ µg/ml) were identified (Supplementary Fig. 1a). The results from BLI analysis (Supplementary Fig. 1b) revealed that the majority of the Fab clones had improved binding abilities, and most of them showed slow or no disassociation compared to the parental antibodies. Clones from the NiV41 parental library still exhibited cross-reactivities to NiV and HeV-RBPs. Eight antibody clones from the NiV41 parental library and six from the NiV42 parental library were converted to IgG format. The mature NiV41 series antibodies showed strong cross-reactivities to both NiV and HeV-RBPs, whereas the mature NiV42 series antibodies showed strong binding to NiV-RBP but weak binding to HeV-RBP (Supplementary Fig. 1c). Consistent with the binding experiment, the mature antibodies from NiV41 had the same cross-neutralizing activities against NiV and HeV pseudoviruses, whereas the NiV42 series antibodies were capable of only neutralizing NiV pseudovirus with high potency (Supplementary Fig. 1d). Four antibodies were selected for further evaluation of their efficacies against authentic virus (Fig. 2a). Compared to the parental antibody, the neutralization ability of the mature antibodies was significantly improved (Fig. 2b). Notably, antibody 42-27 showed a 9- and 13-fold improvement in neutralization ability against $NiV_M$ and $NiV_B$ in relation to its parent antibody NiV42. Simultaneously, we also conducted a comparative analysis of m102.4 and found the $IC_{50}$ values to be 0.025 ($NiV_B$), 0.013 ($NiV_M$), and 0.559 µg/ml (HeV). The NiV41-derived antibodies 41-4, 41-6, and 41-9 are a class of broad-spectrum antibodies against HeV and NiV that are comparable to m102.4, while 42-27 is particularly effective in neutralizing NiV. Results from the NiV41-derived antibodies showed an equivalent level of neutralizing activity against both HeV and NiV, with the capacity to neutralize more than 90% of viruses. Nevertheless, m102.4 was found to have a better effect against NiV. However, it is difficult for m102.4 to reach a 90% neutralizing effect against HeV at the $IC_{90}$ concentration of NiV41-derived antibodies.

We tested the binding kinetics of these antibodies to RBPs to determine affinity (Fig. 2c). Consistent with neutralizing data, their Fab formats bound tightly to RBPs, with affinities ranging from 23.4 to 0.8 nM. Results revealed that NiV41-derived antibodies showed an increase in affinity for RBPs, whereas 42-27 maintained its affinity with the maternal antibody but had a greater neutralizing activity.

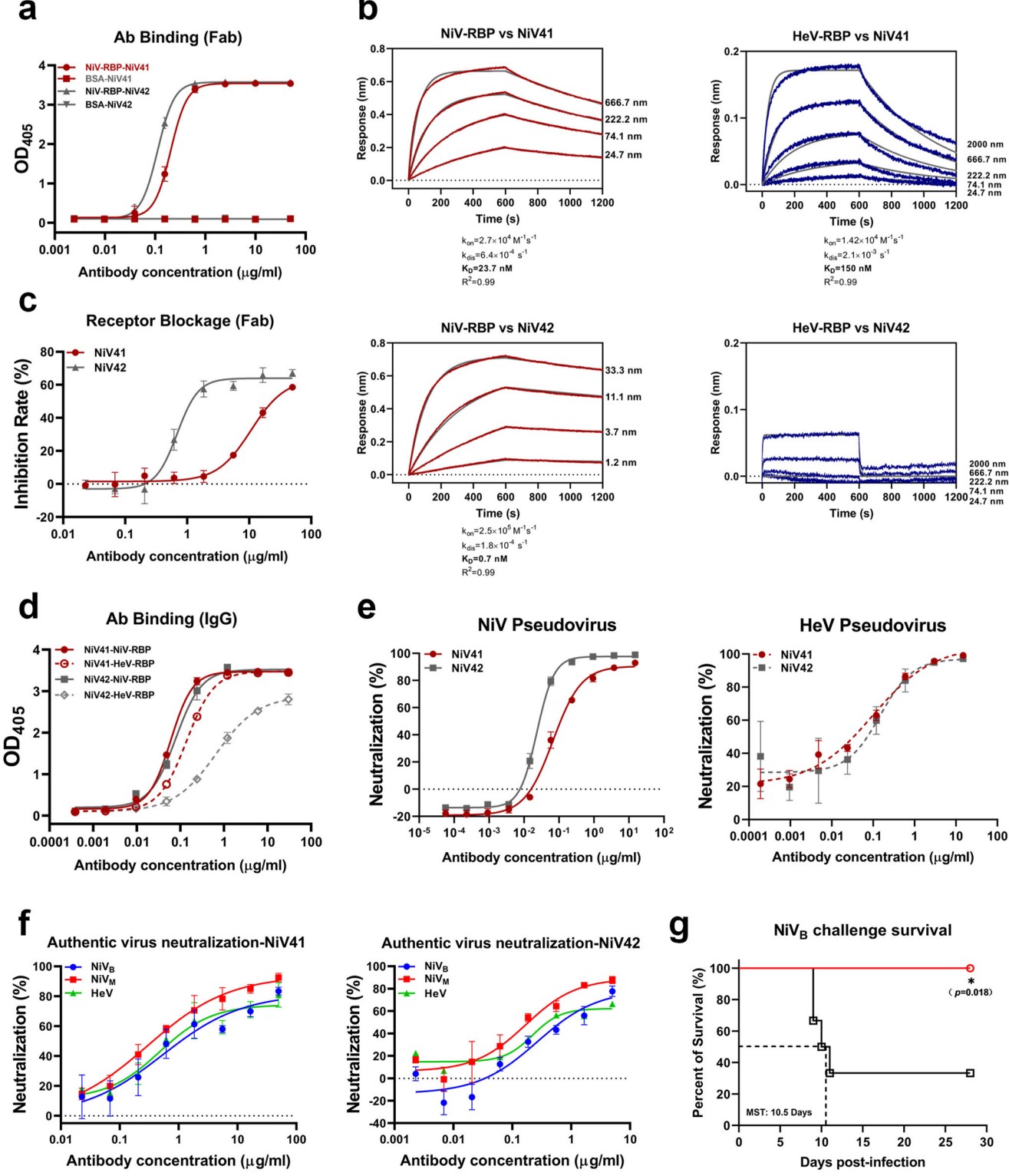

## Identification of parental antibodies and matured antibodies epitopes through alanine scanning

When verifying the operational capacity of antibodies NiV41 and NiV42, it was discovered that both have the ability to impede the binding of the RBP to its receptor at the cellular level. It is believed that the mechanism of action for these antibodies involves blocking receptor binding. As mature antibodies maintain the same heavy chain as their parental antibodies, it is hypothesized that they may share the same mechanism of action and comparable epitopes. To investigate

this, a series of alanine mutants were constructed at key interaction sites between the ephrinB2 receptor protein and RBPs[21]. ELISA was used to assess the binding function of each antibody (Supplementary Fig. 2b, c). The results showed that NiV41 and its mature antibodies had higher sensitivity to the alanine mutants of NiV-RBP than NiV42 and its mature version, particularly at key sites such as C240 and R242, which are located within the central hole of the RBP and are associated with crucial interactions with the ephrinB2 GH loop, as well as the heavy chain complementarity-determining region 3 (CDR3) of m102.4 and

**Fig. 1 | Identification of two neutralizing antibodies from the phage-display library. a** Binding curve of two clones to NiV-RBP. Both FabNiV41 and FabNiV42 could specifically bind to the NiV-RBP. **b** Kinetic binding curves for antibody–antigen interactions were determined by BLI. Biotinylated RBPs were loaded onto probes, while serial dilutions of Fab fragments were then associated with coated probes. $K_{on}$, association rate constant; $k_{dis}$, disassociation rate constant; $K_D$, equilibrium dissociation constant. **c** Blockage of NiV-RBP-Fc binding to Vero cells by Fabs as measured by flow cytometry. Cells were stained with NiV-RBP pre-incubated with NiV41 and NiV42. **d** Binding curve of NiV41 and NiV42 to RBPs. After converted to IgG format, the binding ability of NiV41 and NiV42 to henipavirus RBPs were detected by ELISA. **e** Neutralization activity evolution of antibodies against NiV and HeV pseudovirus. To determine the half-maximal inhibitory concentration (IC$_{50}$), a sigmoidal dose-response curve was applied to the GFP-positive cells or luciferase activities exposed to serially diluted antibody. **f** Neutralization of authentic henipavirus in Vero E6 cells, measured in a plaque reduction neutralization assay. The mixtures of virus and serially diluted NiV41 or NiV42 were added to Vero E6 cells. After 72 h incubation, IC$_{50}$ were calculated by fitting the number of plaques with serially diluted antibody to a sigmoidal dose-response curve. **g** Therapeutic efficacy of NiV41 against a lethal challenge with the NiV$_B$ strain. Hamsters were treated with 3 mg/kg NiV41 or PBS 6 h after intraperitoneal inoculation with NiV. The survival data were collected daily for 28 days after inoculation (day 0). Data ($n = 6$ biologically independent animals per group) were analyzed by the two-sided log-rank (Mantel–Cox) test (*$p < 0.1$). MST, median survival time. Data were represented as the mean ± S.D. from $n = 3$ biologically independent experiments. (**a**, **c**–**f**). Source data are provided as a Source Data file.

HENV26 (Fig. 3a). There are also some differences in sensitive residues among NiV41 and it matured versions possible due to their different light chains. (Fig. 3b).

## Somatic hypermutation in the antibody germline determines neutralizing capacities

Since somatic hypermutation (SHM) is important in antibody maturation, we conducted an immunogenetic analysis of the coding genes of each antibody. Analysis of the nucleotide sequences of the antibody genes revealed that these antibodies retained a high degree of similarity to the germline genes, with the light chain of 42-27 even having a 100% match (Fig. 4a). Both 41-6 and 41-9 share a common origin for their heavy chain and light chain genes, which can be traced back to IGHV3-23*04 and IGLV1-51*01, respectively. The light chain gene of 41-4 originated from IGLV1-44*01, which is the same as that of NiV41. For 42-27, the light and heavy chain genes were derived from IGKV1-39*01 and IGHV1-8*01, respectively. Mutations in the heavy chain gene of 42-27 were mainly in the backbone region, while all three CDR regions retained their original germline characteristics. To assess the impact of the maturation level of the chains on antibody activity, we constructed and expressed chimeric antibodies by inverting the light and heavy chain genes to their respective germline genes. Our protein binding experiments (Fig. 4b and Supplementary Fig. 3a) indicated that when the heavy chain was mutated to an immature gene (germline sequence), the chimeric antibody binding capability was drastically reduced compared to that of fully mature antibodies. However, the light chain germline-reverted antibodies only experienced a slight alteration in their binding activity. Furthermore, our pseudovirus neutralization experiments against NiV demonstrated the same result, indicating that the maturation of the heavy chain was critical for achieving the desired antibody activity (Fig. 4c and Supplementary Fig. 3b).

Studies have previously indicated a correlation between the maturation differentiation of antibodies and their neutralizing capacity[32–35]. Since NiV41 and its mature versions have more SHMs than NiV42 and its mature version, it may be speculated that more SHMs may contribute to the broad-spectrum activity. NiV42 has neutralization activity against HeV. However, after light chain shuffling, the complete germline format 42-27 has significantly increased neutralization activity against NiV but much less neutralization activity against HeV.

## 41-6 protects hamsters against acute infection with Nipah virus

As mentioned above, NiV41 showed therapeutic efficacy in the chronic-infection hamster model. For further evaluation of the antibody's in vivo efficacy, we opted for 41-6 due to its high expression level and performed experiments at a BSL-4 facility to examine its preventive and therapeutic features. We used acute infection hamster models. Prior to infection of hamsters with 1000 LD$_{50}$ NiV$_M$ virus via the IP route, a dosage of 10 mg/kg of antibody was administered, and the body weight and survival status of the animals were monitored daily.

The hamsters that received the antibody were completely protected against viral infection in comparison to the PBS control group (Fig. 5a). We further investigated the efficacy of the antibody by decreasing the dosage to 3 mg/kg, and discovered that its prophylactic effect was dose-dependent, leading to 67% improved survival rate among the treated hamsters (Supplementary Fig. 4a).

In the severe hamster model, unlike intranasal inoculation, intra-peritoneal infection resulted in more severe illness. In this model of severe infection, administering 10 mg/kg of antibody intraperitoneally 3 h after infection resulted in four out of six hamsters surviving (Supplementary Fig. 4b). When the treatment was given twice at 3 h and 3 days postinfection, five of the six hamsters remained alive (Fig. 5b). We also found that only half of hamsters survived when antibody was given on day one and day three after virus challenge (Supplementary Fig. 4c).

## Structure of NiV-RBP in complex with 41-6

To investigate the mechanism of NiV neutralization by 41-6, we determined a cryo-EM structure of the NiV-RBP ectodomain homo-tetramer bound to the Fab fragment of 41-6 at a 2.88 Å overall resolution (Supplementary Fig. 5 and Supplementary Table 1).

In the complex structure, we noticed that the NiV-RBP tetramer was bound to three Fabs (Supplementary Fig. 5a, b). However, due to the flexibility and physiological nature of the viral-host infection, we primarily analyzed the structure of one of the bound Fabs (Fig. 6a and Supplementary Fig. 5). The "neck" domain of chain B (bound with 41-6) has a reverse and parallel folding, and subsequently, a linker connects the head domain to expose the binding epitopes facing upward to 41-6. However, the "neck" domain of chain A (unbound with 41-6) is non-antiparallelly connected to the linker and head domains, with epitopes for 41-6 binding facing the opposite direction (Fig. 6b). These binding epitopes, GLY214$^{NiV-RBP}$, ILE237$^{NiV-RBP}$-VAL244$^{NiV-RBP}$and ILE588$^{NiV-RBP}$, are blocked by the protomer neck domain of chain B (especially from residues 154 to 175) (Fig. 6b). There are some notable different regions in the trajectory of the Cα of the residues involved in the interface of NiV-RBP dimerization and NiV-RBP binding with 41-6 (RMSD >3 Å), including G214, I237, G238, S239, C240, S241, R242, G243, V244, Y581, I588 involved in NiV-RBP binding with 41-6 and S204, P208, V210, G211, Q212, S213, G238, S239, C240, R242, G243, V244, N585, N589 in NiV-RBP dimerization (Fig. 6c). However, their overall conformation is similar (RMSD ≈ 0.59 Å). Further, we compared chain B (bound with 41-6) with the apo monomer NiV-RBP (Supplementary Fig. 6a, PDB 3D11), which may indicate the changes generated from dimerization or 41-6 binding. We also compared the NiV-RBP ectodomain tetramer bound with 41-6 and nAH1.3 (Supplementary Fig. 6a, b PDB 7TXZ). The overall head structures were similar, although there were some regions that deviated, possibly due to the allosteric effects induced by 41-6. These regions included residues 237–244 and 210–214 (Supplementary Fig. 6a, b).

The 41-6 heavy chain binds to NiV-RBP with 960 Å$^2$ of surface area, and the light chain buries 113 Å$^2$, resulting in a total buried

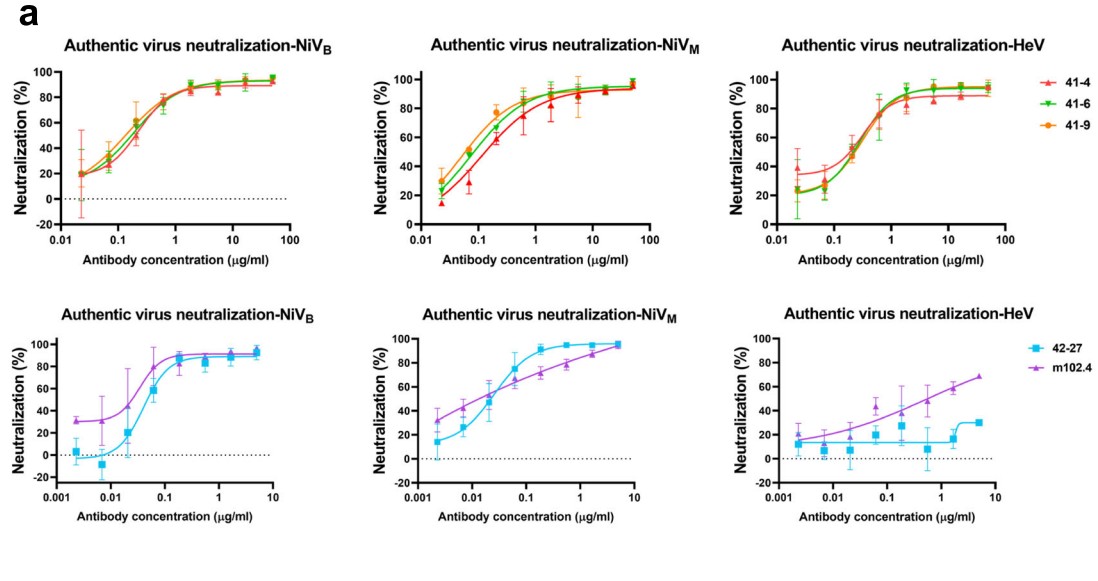

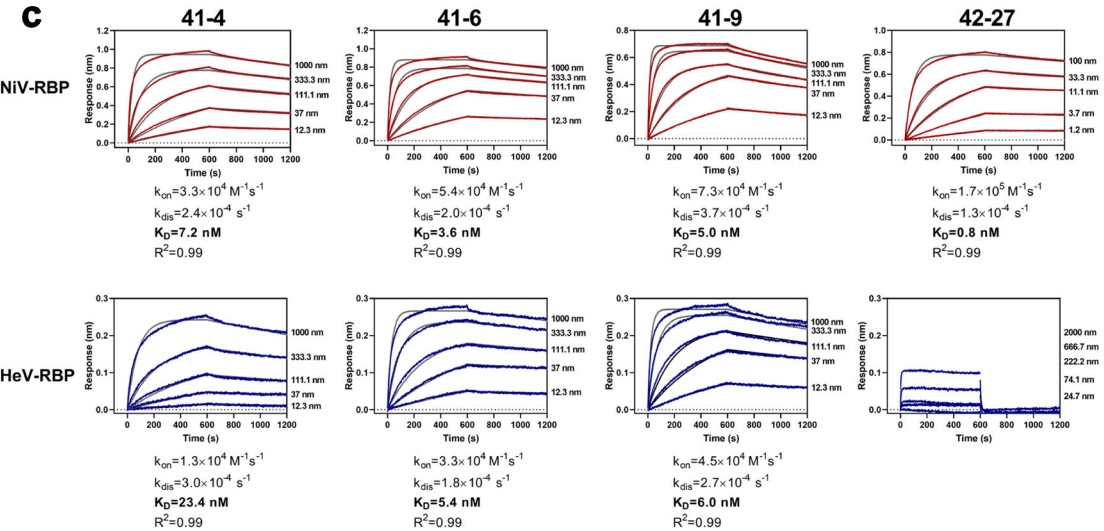

**Fig. 2 | Isolation and characterization of a panel of potent neutralizing antibodies through affinity maturation. a** Neutralization profiles of the authentic henipavirus in Vero E6 cells for antibodies. The mixtures of virus and serially diluted antibodies were added to cells. After 72 h incubation, IC$_{50}$ were calculated by fitting the number of plaques with serially diluted antibody to a sigmoidal dose-response curve. Data are represented as the mean ± S.D. from $n = 3$ biologically independent experiments. **b** Summary of the IC$_{50}$ with 95% confidence intervals of antibodies evaluated by the plaque reduction neutralization titer (PRNT). >5 represents that the antibody has not reached its maximum neutralization effect at the highest dilution concentration. **c** Affinity measurement of antibodies for binding to immobilized NiV-RBP and HeV-RBP, measured by using BLI. The biosensors loaded with Fab fragments (NiV42 or 42-27) or RBPs were shown in Supplementary Fig. 2a. Source data are provided as a Source Data file.

surface area of 1073 Å$^2$ (Fig. 6d). 41-6 mainly utilizes the heavy chain variable region, especially its CDRs, to form interactions with NiV-RBP, including three hydrogen bonds (TYR33$^{CDRH1}$-SER239$^{NiV-RBP}$, SER57$^{CDRH2}$-ILE237$^{NiV-RBP}$, and TYR59$^{CDRH2}$-ARG242$^{NiV-RBP}$ (Fig. 6f).

Besides, the CDRH3 region is likely to be a key area involved in the binding with NiV-RBP, with multiple hydrogen bonds formed (TYR116$^{CDRH3}$-SER242$^{NiV-RBP}$, PRO107$^{CDRH3}$-GLY506$^{NiV-RBP}$, and GLU101$^{CDRH3}$-SER239$^{NiV-RBP}$ SER241$^{NiV-RBP}$, ARG242$^{NiV-RBP}$) (Fig. 6f).

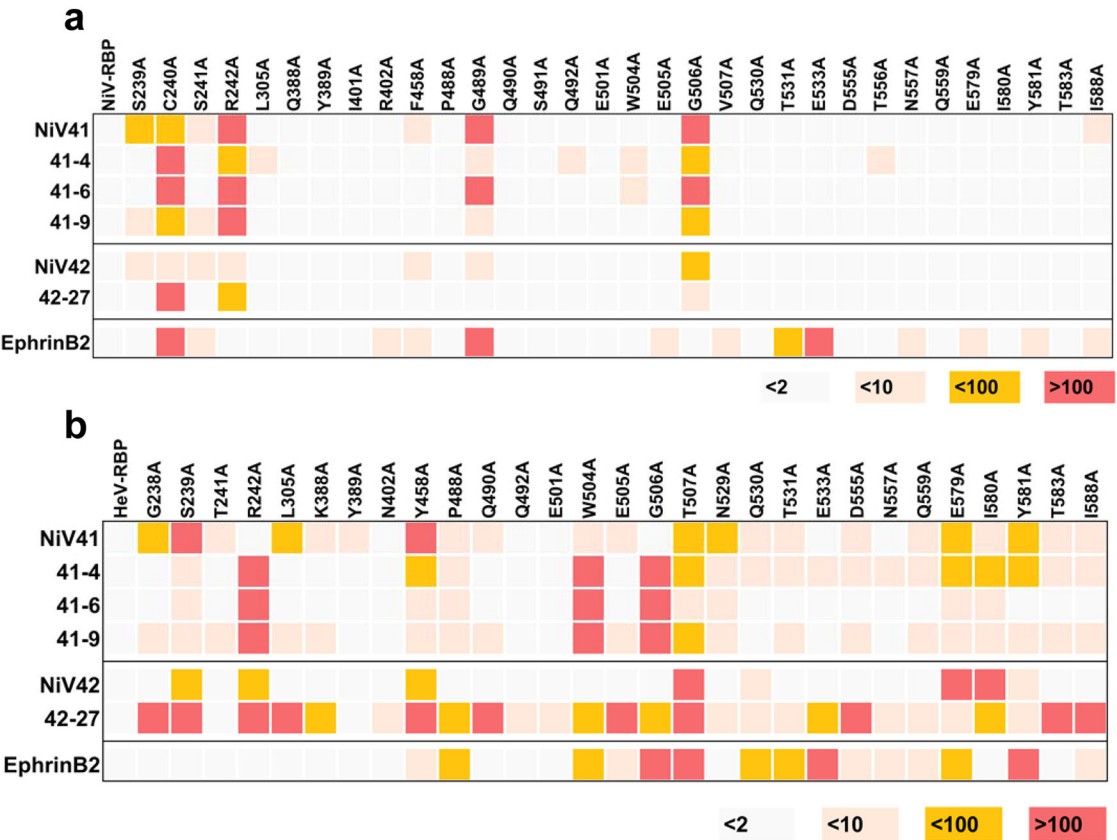

**Fig. 3 | Comparison of the binding site of candidate antibodies and EphrinB2.
a, b** ELISA-based epitope mapping of mAbs with alanine scanned mutants. Utilizing ELISA, 32 mutants of NiV-RBP and 29 mutants of HeV-RBP were employed to evaluate the capacity of antibodies to bind, and the median effective concentration ($EC_{50}$) was calculated by fitting to a four-parameter curve using Prism software. Heatmap shows the fold-change in $EC_{50}$ binding titers of antibodies binding to henipavirus RBPs alanine mutants of compared with the wild-type proteins. The results are color-coded according to the ranges listed (also see Supplementary Fig. 2b, c). Source data are provided as a Source Data file.

Additionally, VAL105^CDRH3, PRO107^CDRH3, and PRO109^CDRH3 insert into 3 hydrophobic pockets, respectively (Fig. 6e–g). VAL105^CDRH3 binds into pocket1 which is composed of ALA558, ASN557, ILE588, and TYR581. PRO107^CDRH3 inserts into pocket2 via hydrophilic and hydrophobic interactions with ALA532, PRO488, VAL507, THR531, and GLY506, GLY489, GLN490, and GLN530. Pocket 3 for PRO109^CDRH3 is formed by PHE458, TRP504, and GLY506. These epitope residues are important for the binding of 41-6 and are consistent with the results of alanine scanning experiments.

The 41-6 epitope is highly conserved between NiV and HeV, thus explaining the antibody's ability to exhibit cross-neutralization activity (Supplementary Fig. 6c). Moreover, we analyze the effect of oligosaccharides on the 41-6 binding to NiV-RBP by docking six reported N-linked glycans into our model (Supplementary Fig. 6d). N378, N417, N481 and N529, four glycans in the head domain, were located away from 41-6-bound epitopes. Even N306, located close to the 41-6 light chain at a minimum average distance of 3.4 Å, also doesn't have a considerable effect on the neutralization activity of 41-6, which may indicate that 41-6 could avoid immune escape.

Our structural analysis of the complex revealed that 41-6 interacts with the RBP mainly via the insertion of its CDRH3 into the pockets in the central cavity, a mode of binding similar to that of receptor proteins. This result is in line with the competitive ELISA results (Supplementary Fig. 7). The comparison of the structures of NiV-RBP with 41-6 and EphrinB2 (EFNB2) demonstrated a considerable degree of similarity in their binding sites (Supplementary Fig. 8d). The CDRH3 of 41-6 was observed to bear a resemblance to the GH loop of EFNB2, both in shape and insertion angle into the hydrophobic pocket of NiV-RBP

(Supplementary Fig. 8a, b). There was little difference in the overall structure between 41-6-bound RBP and ENFB2-bound RBP (Supplementary Fig. 8c). Therefore, 41-6 was an antibody targeting receptor epitopes, which probably mimicked receptor binding to NiV-RBP homotetramer (Supplementary Fig. 8e, f). Furthermore, comparison analysis of henipavirus RBPs bound with m102.3 or 41-6 revealed highly overlapped epitopes, mainly contributed by the Fab heavy chain, especially CDRH3 (Supplementary Fig. 9a–c). CDRH3^m102.3 and CDRH3^41-6 bind to exactly the same hydrophobic cavities (Supplementary Fig. 9b). m102.4 is a matured antibody that shares the same parental antibody clone with m102.3[36]. Computer simulations[37] indicated that their structure is very similar. After comparison of the structure of the complexes of m102.3/RBP and 41-6/RBP, we can conclude that 41-6 is also a potential therapeutic antibody like m102.4 targeting receptor epitopes. When comparing non-receptor binding epitopes on NiV-RBP, it is seen that nAH1.3-Fab[18] (PDB 7TXZ) binds to opposite epitopes on NiV-RBP compared to 41-6 (Supplementary Fig. 9d). And structure superimposition of the NiV-RBP ectodomain complex bound with 41-6 and nAH1.3 shows that the overall conformation of the NiV-RBP tetramer is similar, including head domain, but there is some difference in position of head domain regulated by its linker connected by stalk (Supplementary Fig. 9e).

## Discussion

Antibody therapies for treating respiratory syncytial virus (RSV), SARS-CoV-2, and EBOV have been approved for clinical use. To date, no antibody has been approved for the treatment of NiV infection on the market. Here, we have isolated two monoclonal antibodies, NiV41 and

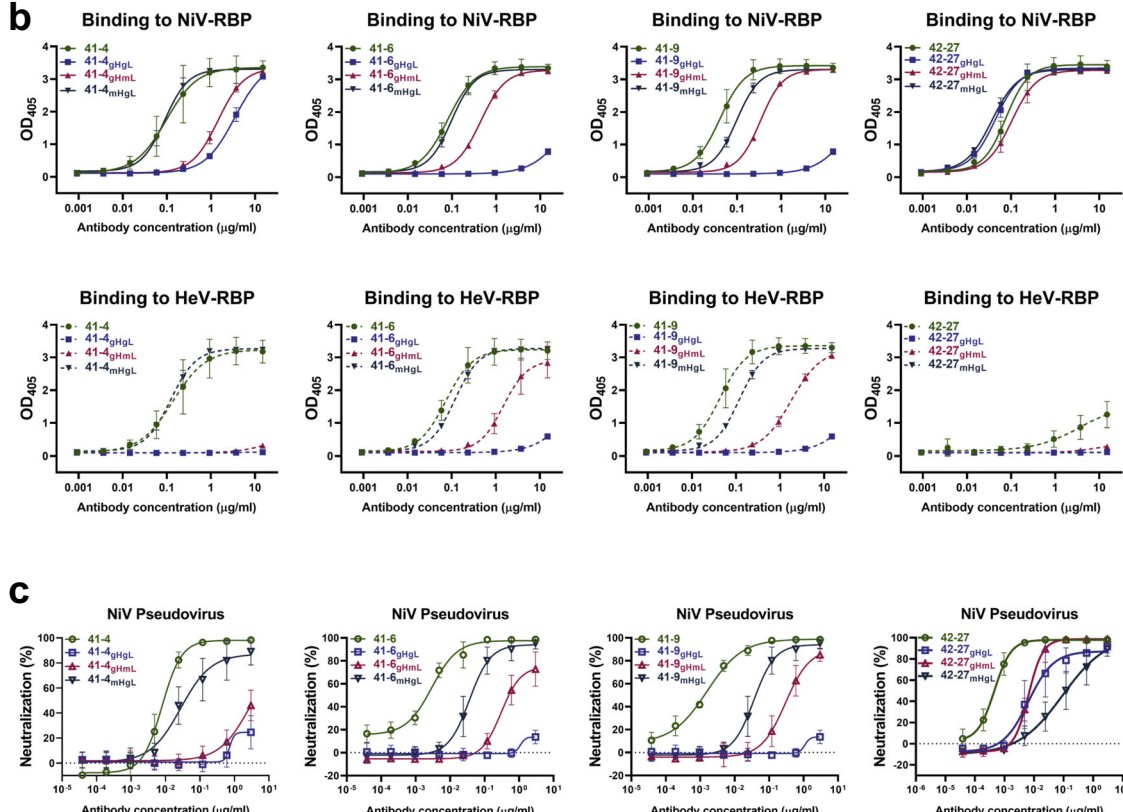

**a**

| mAbs | VH gene | DH gene | JH gene | Identity (HCDR1) | Identity (HCDR2) | Identity (HCDR3) | Identity (nt, VH) | HCDR3 |
|---|---|---|---|---|---|---|---|---|
| NiV41 | IGHV3-23*04 | IGHD2-2*01 | IGHJ6*03 | 70.8% | 100% | 87.5% | 97% | ARDREDIVVVPAP RGYYYYYYMDV |
| 41-4 | IGHV3-23*04 | IGHD2-2*01 | IGHJ6*03 | 70.8% | 100% | 87.5% | 97% | ARDREDIVVVPAP RGYYYYYYMDV |
| 41-6 | IGHV3-23*04 | IGHD2-2*01 | IGHJ6*03 | 70.8% | 100% | 87.5% | 97% | ARDREDIVVVPAP RGYYYYYYMDV |
| 41-9 | IGHV3-23*04 | IGHD2-2*01 | IGHJ6*03 | 70.8% | 100% | 87.5% | 97% | ARDREDIVVVPAP RGYYYYYYMDV |
| NiV42 | IGHV1-8*01 | IGHD2-2*01 | IGHJ6*03 | 100% | 100% | 100% | 95.9% | ARDRGKLVPAAT YYYYYYMDV |
| 42-27 | IGHV1-8*01 | IGHD2-2*01 | IGHJ6*03 | 100% | 100% | 100% | 95.9% | ARDRGKLVPAAT YYYYYYMDV |

| mAbs | VL gene | JL gene | Identity (LCDR1) | Identity (LCDR2) | Identity (LCDR3) | Identity (nt, VL) | LCDR3 |
|---|---|---|---|---|---|---|---|
| NiV41 | IGLV1-44*01 | IGLJ3*02 | 95.8% | 88.9% | 96.3% | 97.3% | ATWDDSLNGWV |
| 41-4 | IGLV1-44*01 | IGLJ3*02 | 79.2% | 55.6% | 96.2% | 92.2% | ATWDDSLNGWV |
| 41-6 | IGLV1-51*01 | IGLJ3*02 | 91.7% | 100% | 77.8% | 91.5% | ATWDDSLHAWV |
| 41-9 | IGLV1-51*01 | IGLJ3*02 | 95.8% | 88.9% | 76% | 92.5% | ASWDDRLNGWV |
| NiV42 | IGKV1-39*01 | IGKJ4*01 | 77.8% | 100% | 100% | 94.2% | QQSYSTPLT |
| 42-27 | IGKV1-39*01 | IGKJ1*01 | 100% | 100% | 100% | 100% | QQSYSTLWT |

NiV42, by panning against NiV-RBP. NiV41 and NiV42 were subjected to further maturation.

The HeV-RBP contains 604 amino acids in length and has 83% similarity to NiV[38]. Although NiV and HeV are both members of the Henipavirus genus and share a high degree of similarity, the neutralizing antibodies available against the RBPs of these two viruses exhibit distinct characteristics that can be classified into three types,

including cross-neutralizing antibodies as well as specific neutralizing antibodies against either NiV or HeV[29,39]. We found that the antibodies derived from NiV41 (e.g., 41-6) exhibit cross-neutralizing activities against NiV and HeV. In contrast, those derived from NiV42 (e.g., 42-27) show specific NiV neutralization. We also noticed that 42-27 was a germline antibody, while 41-6 had more SHMs. This suggests that SHM might aid in the development of cross-reactivity. Presently, the

**Fig. 4 | Somatic hypermutation in the antibody germline determines neutralizing capacities. a** Summary of the germline gene (V-D-J for heavy chain and V-J for light chain), V-gene nucleotide somatic hypermutations, and CDR3 of antibodies. The analysis of the germline nucleic acid sequence of antibodies was conducted using IMGT/V-QUEST. **b** Binding curve of germline-reverted antibodies to RBPs. The binding results were analyzed by fitting to a four-parameter curve using GraphPad Prism software. **c** Neutralization activity evolution of germline-reverted antibodies against NiV pseudovirus. The neutralization percentage of NiV pseudovirus relative to non-antibody-treated controls was determined by counting the

number of GFP-positive cells 24 h postinfection. $IC_{50}$ values were then calculated through nonlinear regression analysis using Prism. gHgL, VH germline paired with VL germline. gHmL, VH germline paired with antibody VL. mHgL, antibody VH paired with VL germline. The antibodies 41-6gHgL and 41-9gHgL were identical and labeled differently in the illustration to facilitate comparison. This same principle applied to 41-6mHgL and 41-9mHgL. Data were represented as the mean ± S.D from $n = 3$ biologically independent experiments. (**b**, **c**). Source data are provided as a Source Data file.

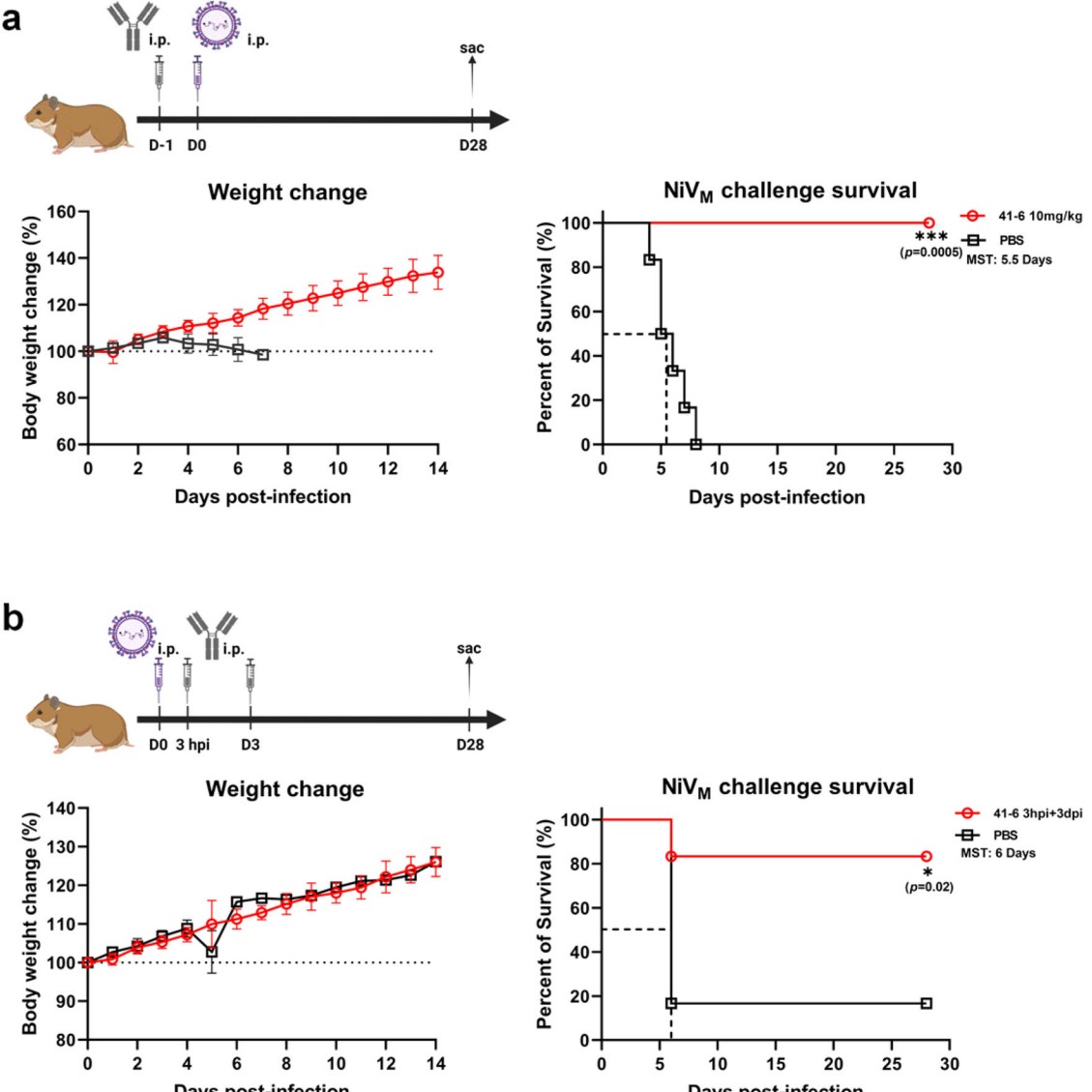

**Fig. 5 | 41-6 protects hamsters against acute infection with Nipah virus.**
**a** Prophylactic efficacy of 41-6 against a lethal challenge with the NiV_M strain. Antibodies diluted in PBS were administered at 10 mg/kg through IP 24 h before viral infection. Control group animals received the same volume of PBS. Hamsters were challenged with a virus by IP route (day 0). The weight change was collected daily for 14 days after inoculation and the survival was observed for a duration of 28 days. **b** Therapeutic efficacy of 41-6 against a lethal challenge with the NiV_M strain. Hamsters were treated with PBS buffer or 10 mg/kg 41-6 for 3 h and 3 days

after intraperitoneal inoculation with the NiV virus. The weight change was collected daily for 14 days after inoculation and the survival was observed for a duration of 28 days. Error bars represent the mean ± S.D. Data ($n = 6$ biologically independent animals per group) were analyzed by the two-sided log-rank (Mantel–Cox) test using Prism software (*$p < 0.1$, ***$p < 0.001$) (see also Supplementary Fig. 4). MST median survival time. The schemes were created with BioRender.com. The PBS-treated animals in **b** are identical to those in Supplementary Fig. 4b, c. Source data are provided as a Source Data file.

identified neutralizing antibodies targeting the RBP have exhibited cross-reactivity against henipavirus[29,36,39]. Although certain antibodies have been isolated from donors who have received the HeV vaccine, most of these antibodies specifically target the HeV[29,39]. Our research has revealed that 42-27 and its germline antibody are highly specific

and efficacious antibodies against NiV with no observed poly specificity. As a result of five mutations occurring within the germline VL genes, NiV42 has acquired the capacity to bind to the HeV. Several studies on HIV germline antibodies have indicated that a moderate level of somatic hypermutation is necessary for the breadth of

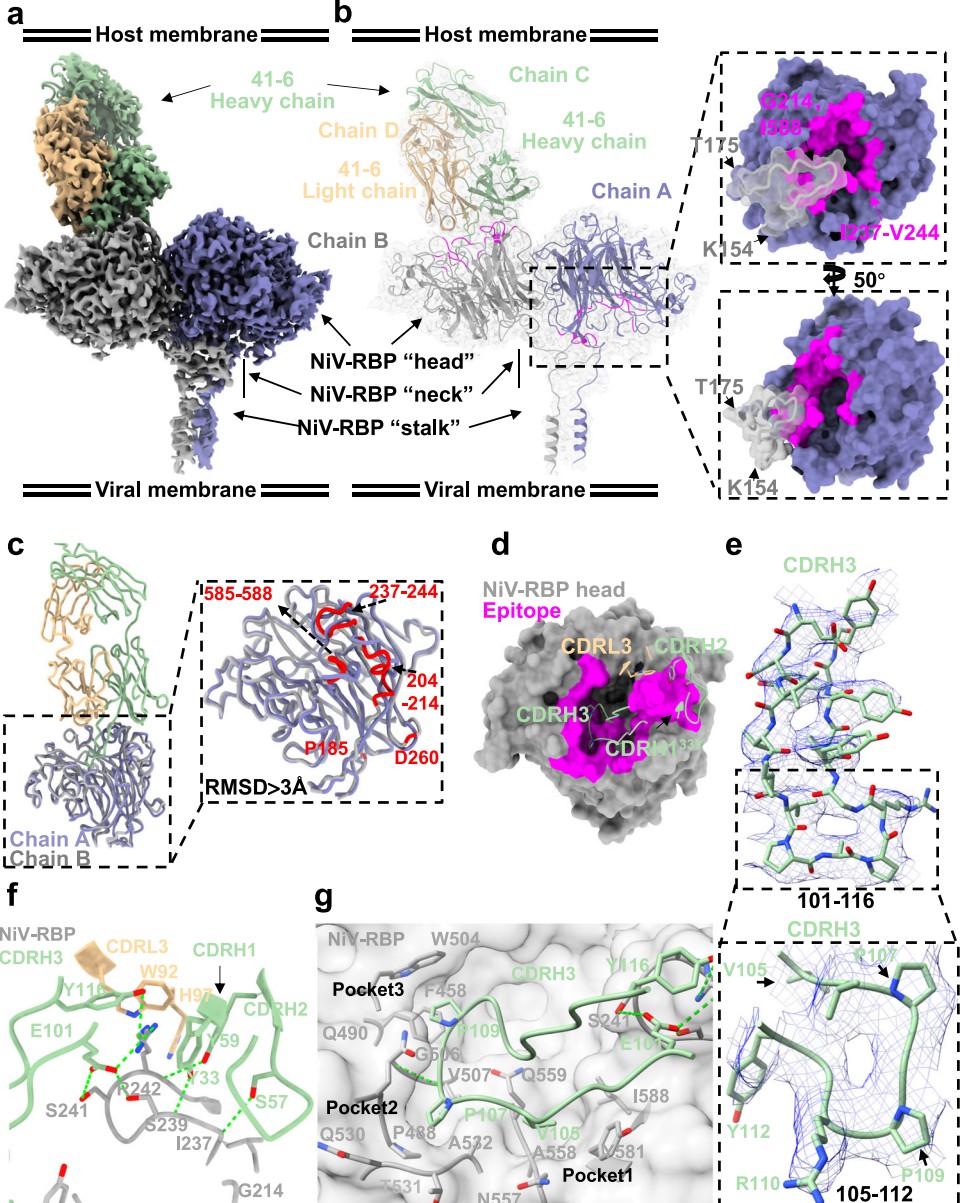

**Fig. 6 | Structure of NiV-RBP in complex with the 41-6 neutralizing Ab. a** Cryo-EM structure of the NiV-RBP ectodomain in complex with the neutralizing antibody 41-6 Fab fragment (41-6) represented by Cryo-EM map. Each of the NiV-RBP protomers is shown in a different color. 41-6 heavy and light chains are green and tan, respectively. **b** Interface of NiV-RBP with 41-6 in complex, which is shown as a model fitted into the density map. Left, interaction epitope between NiV-RBP protomers and 41-6. The epitope of NiV-RBP (Chain A) presumed by docking into NiV-RBP with the 41-6 complex is shown in magenta. Right, a zoomed-in view of the interaction between NiV-RBP dimers. **c** Superimposition of the NiV-RBP protomer with and without 41-6 from the above complex, shown as colored ribbon diagrams, NiV-RBP protomer (gray) with 41-6 indicated Chain B, adjacent protomer (medium slate blue) without 41-6 indicated Chain A. Right, zoomed-in comparison of Cα trace of NiV-RBP head between Chain A and Chain B. Regions with the highest deviations (RMSD >3 Å) between Chain A and Chain B are colored red. **d** Interaction analysis

between the NiV-RBP head and 41-6. NiV-RBPs are shown as molecular surfaces. 41-6-VH: CDR1, CDR2, CDR3; 41-6-VL: CDR1, CDR2, CDR3, and FR-L3, which from each Fab are shown as different colored as ribbon diagrams. **e** Zoomed-in view of 41-6 CDRH3 densities map fit with a stick model, shown key residues binding with NiV-RBP, three residues (V105, P107, P109) insert into different pockets, respectively. **f** Zoomed-in view of interaction analysis of NiV-RBP with 41-6. 41-6 and NiV-RBP are shown as ribbon diagrams, colored green (heavy chain), tan (light chain), and gray, respectively. Critical contact residues are shown as sticks. Oxygen and nitrogen atoms are colored red and blue, respectively. Hydrogen bonds are shown as green dotted lines. **g** Interaction of NiV-RBP with 41-6-VH-CDR3. NiV-RBP is shown as a transparent molecular surface colored by gray, 41-6-VH-CDR3 is shown as a ribbon, and critical contact residues are shown as sticks. Oxygen and nitrogen atoms are colored red and blue, respectively. Hydrogen bonds are shown as green dotted lines.

neutralizing antibodies targeting the V1/V2 epitope[34]. Similarly, our research suggests that germline antibodies may be responsible for the antibody response to henipavirus antigens. We also uncovered antibodies generated through light chain shuffling from NiV41 had a more powerful neutralizing capability, and the maturity of their light chains was slightly advanced compared to the parental antibodies. These results implied somatic hypermutation produces iterative affinity

selection and maturation that provides the increasing affinity for antibody recognition of antigens.

Original antigenic sin (OAS) is one of the key factors that restricts the efficacy of vaccines[40,41]. Despite the high homology between the NiV and the HeV, there is a certain degree of immunogenic bias in vaccine-induced humoral immunity. The identification of specific germline antibodies is expected to reveal variations in RBPs epitopes

between the two viruses, which can aid in the development of germline-targeting universal vaccines against henipaviruses. Individuals who have received immunization with HeV-RBP tend to produce antibodies that demonstrate cross-reactivity; however, some of them still display exclusive specificity toward HeV without response to NiV. This could reduce the vaccine's broad-spectrum protective capabilities. The degree to which the selection of a particular target protein dictates the spectrum of neutralizing antibodies is still not entirely clear. Additionally, there is a lack of adequate research regarding the frequency of genes and the maturation principles associated with antibodies that target the NiV. Understanding the immunodominance landscape of the RBP will identify the requirements for a broader henipavirus antibody response and provide the foundation for developing broad-spectrum vaccines.

Our structural data revealed two populations where the soluble NiV-RBP ectodomain, that has two heads near the viral terminus and two heads near the host terminus, was bound with either one or three 41-6 Fabs. Given resolution limitations and the structural basis of physiological viral-host infection, we resolved a Cryo-EM of a single antibody (41-6 Fab)-bound homotetramer NiV-RBP, which showed that the two head regions distant from the viral membrane form an asymmetric dimer. It was observed that one of the four head domains directs its receptor binding site towards the host cell membrane, while the remaining three sites are oriented towards the viral membrane. This finding is in line with the previous study[18]. In contrast to the non-receptor-competitive antibody nAH1.3, our 41-6, a receptor-competitive antibody, is limited to binding to only one of the two head regions located near the host terminus. This is possibly because the binding epitope in the other protomer is obstructed by the adjacent subunit. This type of binding has not been previously documented. Upon further investigation, we found the epitope residues of 41-6 partially overlap with the receptor binding sites in RBP[21] and are similar to those of other receptor-like antibodies[22]. It was also disclosed that the binding of 41-6 primarily induces a reconfiguration of the stalk and neck domains. The antigen interaction of 41-6 and m102.3 showed a high level of similarity due to their shared long CDRH3 length and conformation. However, the light chain differences between the two antibodies resulted in different angles of entry into the central cavity. Notably, the coding germline gene segment for 41-6 was IGHV3-23, which was identical to that of HENV26, while the coding germline gene segment for m102.3 was IGHV1-69, a gene that is found at a high frequency in the human repertoire and is often observed in the early immune response to a variety of pathogens[42]. A previous study indicated that the IGHV3 gene is the most stable germline antibody family for human VH domains[43]. Additionally, sequences in CDRH1 and CDRH2 of 41-6 and HENV26 had a considerable degree of similarity. The germline amino acid 59Y, located on IGHV3-23, was found to be involved in the interaction with the antigen. Investigating and identifying more human monoclonal antibodies will help us gain a deeper understanding of the body's immune response to henipaviral infection.

The efficacy of therapeutic drugs and vaccines against henipavirus has been tested using various animal models. Research has demonstrated the protective effects of m102.4 on ferrets against NiV, indicating its effectiveness as a therapeutic agent[44]. Additionally, Africa green monkeys (AGMs) have also been used to demonstrate the protective effects of m102.4 against both NiV[45] and HeV[46]. Furthermore, studies on ferrets with HENV26 and HENV32 have confirmed their protective efficacy against NiV[29]. The golden hamster, in which researchers can replicate severe respiratory infections and encephalitis in humans, is often used in NiV research[47–49]. Compared to ferrets and AGMs, hamsters necessitate a much higher challenge dose to cause lethal disease, leading to a more rapid progression of the illness and a shorter treatment period[39]. NiV spreads through the respiratory tract and bodily fluids, and unlike other research that used nasal

inoculation, our study employed intraperitoneal injection. Previous studies have suggested that this method of injection can quickly initiate infectious diseases, leading to severe symptoms in the hamster model[47]. An evaluation of the in vivo efficacy of antibodies against $NiV_B$ revealed that the mortality rate in the control group was 60%. Although the Bangladesh strain had more virulence when infecting people, its infectivity in a hamster model was weaker than that of the Malaysia strain, resulting in a less severe disease[30,50]. In subsequent animal experiments, we selected the Malaysia strain to evaluate the activity of 41-6 and observed a different mortality rate in the control group. The findings indicate that NiV41 was able to provide protection in the chronic-infection hamster model, while 41-6 was effective in protecting against severe infection; however, importantly, variations in animal sources and virus storage cannot be completely disregarded, and discrepancies may be seen due to experiments being conducted in different laboratories. Our research has shown that 41-6 is capable of providing complete prophylactic protection to hamsters infected with NiV. Therefore, it could be used as a preventive drug during NiV pandemic. We also showed that 41-6 can be effective in the initial exposure. Then it could be used as a postexposure prophylactic treatment for high-risk populations. More animal studies are required to verify more clinical potentials of 41-6. For NiV receptor-like antibodies, m102.4 has been shown to be protective in ferrets and AGMs, while HENV26 has been demonstrated to be efficacious in ferrets, and 41-6 has been found to be protective in hamsters. To gain a better understanding of the differences in their efficacy in vivo, a more thorough comparison under the same experimental conditions is necessary. In summary, 41-6 is a promising antibody-based treatment that could be used clinically to treat NiV infection.

## Methods

### Ethics statement
Hamster studies were approved by the Life Science Ethics Committee of the Wuhan Institute of Virology, Chinese Academy of Sciences. (approval no. WIVA45202306).

### Cells and viruses
Vero cells (catalog no. GDC0029), Vero E6 cells (catalog no. GDC0146), and 293T cells (catalog no. GDC0187) were obtained from the China Center for Type Culture Collection (CCTCC). These cells were cultured in Dulbecco's modified Eagle's medium (DMEM, Gibco) supplemented with 10% fetal bovine serum (FBS, Gibco) and penicillin–streptomycin (Beyotime) in a 37 °C, 5% $CO_2$ atmosphere. FreeStyle™ 293-F cells (cat#R79007) were obtained from Thermo Fisher and maintained in Freestyle™ 293 expression medium (Thermo Fisher) at 37 °C in an 8% $CO_2$ atmosphere.

Nipah virus Malaysia strain (CSTR: 16533.06.IVCAS 6.7489), Nipah virus Bangladesh strain (CSTR: 16533.06.IVCAS 6.7488), and Hendra virus (CSTR: 16533.06.IVCAS 6.7487) were obtained from the National Virus Resource Center, Wuhan Institute of Virology, Chinese Academy of Science. Each virus was passaged in Vero cells. All processes in this study involving authentic henipavirus were performed at the Wuhan National Biosafety Laboratory under biosafety level 4 (BSL-4) conditions. Virus titers were determined using 50% tissue culture infectious dose ($TCID_{50}$) method. The median lethal dose ($LD_{50}$) was calculated by Reed and Muench.

### Expression and purification of henipavirus RBPs
The wild-type sequence of the $NiV_M$ RBP protein (GenBank:NC_002728.1, residues 183-602) was cloned into a pCAGGS expression vector, with a human IgG Fc fragment at the C-terminus. Subsequently, the plasmid containing the NiV-RBP-Fc fragment was verified by DNA sequencing. HEK293F cells were then transfected with plasmid using polyethyleneimine (PEI-25 kDa, Polysciences). After 5 days of culture, the supernatant was collected, and the soluble

protein was purified by protein A resin (GE Healthcare). Moreover, the global head domain of NiV-RBP in pCAGGS with a His-tag at the C-terminus was expressed and purified from HEK293F cells through Ni Sepharose (GE Healthcare). Additionally, the sequence encoding the HeV-RBP global head domain (GenBank:NC_001906.3, residues 183-602) was inserted into the expression vector pSectag2A, with a His-tag at the C-terminus.

## Antibody screening from the phage library
A large, naïve human phage display Fab library was constructed in-house and panned. Phages displaying NiV-specific Fabs were enriched after multiple rounds of biopanning on immobilized NiV-RBP-Fc in one well of a microtiter plate. For each panning round, a BSA-coated well was used beforehand to deplete unspecific phages. The wells were washed three times with phosphate-buffered saline (PBS) plus 0.05% Tween-20 (PBST) and blocked with 3% milk in PBS. Approximately $10^{11}$ phages were added to the coated wells and incubated for one hour at 37 °C. Unbound phages were removed by washing with PBST (ten times in the first round and five times added per round thereafter). The retained phages were amplified in exponentially growing E. coli TG1 cells, infected with M13K07 helper phages, and purified for the next round of selection. Enrichment after each panning round was determined by infecting TG1 cells with tenfold serial dilutions, after which the bacteria were plated on 2× YT agar plates with 100 μg/ml ampicillin. The antigen reactivity of each round was determined by polyclonal phage ELISA. Several individual colonies isolated after the last round of panning underwent monoclonal phage ELISA to select positive binders. The DNA of the selected binders was obtained by plasmid extraction and determined by sequencing.

Using the genes from the naïve Fab library, two chain shuffling libraries were constructed for antibody affinity maturation. The genes encoding the Fd (VH + CH1) fragments of NiV41 and NiV42 were amplified and then fused to the VL repertoires through splicing by overlapping extension PCR. The resulting fragments were digested with sfiIand purified via gel electrophoresis before being ligated into the digested backbone vector to generate the light chain shuffled library. Subsequently, panning was carried out according to the established protocol.

## Expression and purification of Fabs
After screening, the DNA fragment of the selected clones was inserted into the modified pComb3XSS vector with a C-terminus 6× His-tag and Flag-tag. To express the Fab, the vector was transformed into E. coli HB2151 cells, which were grown at 37 °C in SB medium to an optical density of A600 -0.6–0.8. Expression was induced with isopropyl-1-thio-β-D-glucopyranoside (IPTG) (1 mM) at 30 °C for 12–16 h. To prepare the periplasmic extract, the bacterial cells were pelleted and resuspended in PBS. Polymyxin B sulfate (Sigma–Aldrich, 0.5 mu/ml) was added to the suspension at a ratio of 1:1000 (volume of polymyxin B: culture volume). After 1 h of incubation at room temperature, followed by centrifugation, the cell lysate was collected, and the Fab was purified using Ni-NTA spin (QIAGEN). For protein production measurement by cryo-electron microscope analysis, after the Ni-NTA spin purification, the Fab protein was further purified by protein G resin (GE Healthcare).

## Construction and production of IgG
For the conversion and preparation of IgG, selected Fab clones were converted to IgG1 formats. The intact sequence of the Fab clone was fused with the human IgG1 Fc fragment and was then cloned into the vector pVitro2-neo-mcs (InvivoGen). IgG was produced by transient expression in HEK293F cells and purified with a protein A resin.

## Enzyme-linked immunosorbent assay (ELISA)
High-binding 96-well plates (Corning) were coated with recombinant NiV-RBP-Fc, NiV-RBP, or HeV-RBP at 4 μg/ml and left to sit overnight at 4 °C. Subsequently, the plates were blocked with PBS containing 3% skim milk (w/v, Bio-Rad) at 37 °C for 1 h. The plates were washed with PBST three times, and polyclonal phages, monoclonal phages, or serially diluted antibodies were added, and plates were incubated at 37 °C for 1.5 h. Next, the plates were washed with PBST five times, and HRP-conjugated anti-M13 (Sino Biological, cat#11973-MM05T-H, 1:3000 dilution), HRP-conjugated anti-Flag (Sigma-Aldrich, cat#A8592, 1:2000 dilution), or HRP-conjugated anti-human IgG Fc antibody (Sigma-Aldrich, cat#A0170, 1:5000 dilution) was used as a secondary antibody. After 1 h of incubation, the plates were washed with PBST five times. The binding was measured with the subsequent addition of substrate diammonium 2,2-azinobis (3-ethylbenzothiazoline-6-sulfonate) (ABTS, Invitrogen), and the absorbance signal at 405 nm was measured with a BioTek Synergy H1 microplate reader (BioTek Instruments). The median effective concentration ($EC_{50}$) was determined by curve fitting using four-parameter nonlinear regression (GraphPad 8.0).

## ELISA-based competition assay
Competitive ELISA experiments were conducted with slight modifications. Diluted competitive antibodies or receptor proteins were mixed with biotinylated antibodies held at a constant concentration of 0.03 μg/ml. After coating with antigen and blocking with milk, the mixture was incubated for 1.5 h, and then the HRP-conjugated anti-streptavidin antibody (Sigma-Aldrich, cat#18152, 1:8000 dilution) was added. The plates were washed, and the absorbance signal was assessed as previously outlined.

## Flow cytometry-based receptor binding inhibition assay
Vero cells grown to subconfluency was detached by sodium citrate and aliquoted into tubes. After washing with PBS + 2%FBS, the cells were blocked with 1% BSA in PBS on ice. Tenfold serial dilutions of Fabs were incubated with NiV-RBP-Fc (at a final concentration of 10 nM) on ice for 1 h, and then the mixture was added to cells for incubation on ice for 1.5 h. Cells incubated with RBP-Fc were used as a positive control. After washing three times with PBS + 2%FBS, the cells were stained with goat anti-human IgG Fc-DyLight650 (Invitrogen, cat#SA5-10137, 1:50 dilution) on ice for 1 h. Following three times washes, the cells were resuspended and analyzed by flow cytometry using a BD LSR flow cytometer (BD Biosciences). The inhibition rate was determined by comparing the mean fluorescence intensity (MFI) of the antibody mixture to that of the positive control. A figure exemplifying the gating strategy is provided in Supplementary Fig. 10.

## Pseudotyped virus neutralization assay
The NiV pseudovirus was generated following established protocols[51], in which the pCAGGS plasmids encoding the NiV-RBP and F genes were cotransfected into subconfluent 293T cells using Lipofectamine™ 3000 (Invitrogen). The cells were cultured at 37 °C with 5% $CO_2$ and infected with of VSV△G-EGFP/VSV G (MOI = 4) for 2 h, followed by washing with PBS three times. DMEM supplemented with 2% FBS was added. The supernatants containing VSV-EGFP-NiV-RBP/F pseudovirus were collected on the following day and filtered through a 0.45-μm filter to remove cell debris. The virus titer was measured in Vero cells.

Generation of HIV-1 pseudotyped with HeV-RBP/F was performed according to Khetawat[52]. Codon-optimized DNA encoding HeV-RBP with a shortened cytoplasmic tail (RBP-CT32) or HeV-F protein lacking the last 25 amino acids (F-522) was synthesized and cloned into the pCAGGS vector. 293 T cells were cotransfected with pCAGGS/HeV-RBP-CT32, pCAGGS/HeV-F-522, and pNL4-3-Luc-R at a ratio of 1:1:7 using Lipofectamine™ 3000. The supernatants containing HIV-Luc-HeV-RBP/F were harvested 72 h post-transfection and passed through a 0.45-μm filter.

Gradient diluted antibodies were incubated with an equal volume of pseudovirus at 37 °C for 1 h. For VSV-EGFP-NiV-RBP/F pseudovirus

neutralization, the mixture was added to a monolayer of Vero cells in a 96-well plate, and the plates were incubated for 24 h. Fluorescence was measured using an Operetta high-content imaging system combined with Harmony imaging and analysis software (PerkinElmer). For HIV-Luc-HeV-RBP/F pseudovirus neutralization, 293T cells seeded into solid white 96-well plates were infected with pretreated pseudovirus particles and incubated for 48 h. The efficiency was assayed for luciferase activity using luciferase substrate on a microplate reader. The inhibitory effects of each dilution were evaluated, and the half-maximal inhibitory concentration ($IC_{50}$) was calculated using Prism software.

### Plaque reduction neutralization assay

Antibodies were diluted threefold in DMEM with 2.5% FBS and then combined with an equivalent volume of virus suspension. Following incubation at 37 °C for 1 h, the mixture was added to monolayered Vero E6 cells in 24-well plates and incubated for an additional hour. Upon removal of the mixture, 0.5 ml of DMEM containing 2.5% FBS and 0.9% methylcellulose (Sigma) was added to each well. The plates were then incubated in a 5% $CO_2$-air incubator at 37 °C for 4–5 days. Subsequently, the plaques were stained with crystal violet and counted 24 h later. The neutralizing titer was determined by calculating the reciprocal of the highest antibody dilution that suppressed 50% of plaque formation. The plaque reduction neutralization titer (PRNT) was calculated using the "inhibitor vs. normalized response (variable slope)" model in GraphPad Prism 8.0 software.

### Biolayer interferometry binding assay

BLI kinetic measurements were acquired at 30 °C with a ForteBio Octet Red™ instrument (Sartorius). To examine the binding of Fabs to different antigens, biotinylated RBPs were prepared according to the EZ-link®Sulfo-NHS-LC-Biotin (Pierce) reagent instructions and were diluted to 35 μg/ml in PBS containing 0.01% Tween-20 and 0.1% BSA. RBPs were immobilized onto streptavidin-coated biosensors and serial dilutions of Fab were then incubated. After subtracting the non-specific binding using a reference biosensor, global data fitting to a 1:1 binding model in ForteBio Data Analysis Software was used to determine values for the $k_{on}$ (association rate constant), $k_{dis}$ (disassociation rate constant), and $K_D$ (equilibrium dissociation constant).

### Alanine scanning mutagenesis to identify Ab-binding residues

Alanine mutants of the NiV and HeV-RBPs were designed according to the crystal structure of the RBP-ephrinB2 complex. PCR-mediated site-directed mutagenesis was used to introduce mutations into cDNAs encoding the NiV-RBP or HeV-RBP globular head domain. The plasmids encoding mutants were transfected into 293 F cells, and proteins were purified by Ni Sepharose as described above. The $EC_{50}$ value was determined using Prism software.

### Immunogenetic analysis

Analysis of the germline nucleic sequence of antibodies was conducted using IMGT/V-QUEST. Germline-reverted antibodies were generated by eliminating any somatic mutations from the inferred germline gene V/D/J sequence. Several antibody variants were created, which included $IgG_{gHgL}$ (germline heavy chain combined with germline light chain), $IgG_{gHmL}$ (germline heavy chain combined with mature light chain), and $IgG_{mHgL}$ (matured heavy chain combined with germline light chain). The binding capacity and neutralizing activity of the purified germline-reverted antibodies were then assessed using ELISA and pseudovirus neutralization assay, respectively. The differences between the parental antibodies and the newly constructed variants were then evaluated using Prism.

### Animal protection assay

Five- to six-week-old female Syrian golden hamsters were randomly allocated into groups. They were kept in SPF animal facilities at the Wuhan Institute of Virology, Chinese Academy of Science. Viral infection was conducted in a BSL-4 facility in accordance with the guidelines for the care and use of laboratory animals and the Institutional Review Board of the Wuhan Institute of Virology, Chinese Academy of Science.

### Prophylactic study

Female hamsters aged between 5–6 weeks were intraperitoneally administered PBS vehicle, 300 μg (≈3 mg/kg), or 1 mg (≈10 mg/kg) antibody 41-6 one day prior to challenge with 1000 $LD_{50}$ of $NiV_M$. The virus was injected intraperitoneally with a stock diluted in PBS containing 2% FBS. The animals were monitored daily for clinical scoring and weight change and euthanized when they reached either the institutionally approved endpoint score criteria or the study endpoint (28 days postinfection).

### Therapeutic study

Female hamsters aged between 5–6 weeks were assigned into groups randomly, and were challenged with $10^5$ $TCID_{50}$ $NiV_B$ by IP route. NiV41 diluted in PBS were administrated through the IP route at 6 h post-infection. Control group animals received the same volume of PBS.

Female hamsters aged between 5–6 weeks were exposed to IP to 1000 $LD_{50}$ of $NiV_M$. Subsequently, 1 mg of antibody 41-6 was administered to the animals 3 h postinfection, or 3 h postinfection and 3 days postinfection, or 1 day postinfection and 3 days postinfection. Additionally, PBS vehicle was administered 1 day postinfection.

### Cryo-EM sample preparation

To prepare NiV-RBP with 41-6 complex purified for cryo-EM analysis, NiV attachment glycoprotein protein ectodomain (NiV-RBP) was incubated with an excess molar ratio of purified 41-6 at 4 °C overnight. The complex was purified through Superose 6 Increase 10/300 GL (GE Healthcare) gel filtration in a buffer containing 50 mM Tris/HCl (pH 8.0), 150 mM NaCl and 2% glycerol.

A drop of 3 μl of 0.16 mg/ml NiV-RBP–41-6 complex was applied onto glow discharged (10 s at 20 mA) copper grid coated with a thin continuous carbon layer (Quantifoil, R2/1 200-mesh) and treated with 0.1% polylysine. The grid was rapidly frozen using vitrobot Mark IV (Thermo Fisher Scientific) using a blot time of 1.5 s at 100% humidity and 4 °C. The quality of the sample was screened using a Glacios cryomicroscope (Thermo Fisher Scientific) operating at 200 kV.

### Data collection and processing

The grid was imaged in a Titan Krios G3 operated at 300 kV (Thermo Fisher Scientific), equipped with a Gatan K3 detector and the Gif Quantum energy filter with 20-eV slit width. A total of 9103 micrographs were automatically acquired with EPU software at a magnification of 81,000 x with a pixel size of 0.82 Å. The defocus range is from −2.0 to −3.2 μm, 30 frames with 1.7 electrons per pixel per frame, a total exposure of 3 s, and an accumulated dose of 51 $e^-/Å^2$.

All micrographs were performed using CryoSPARC 3.2.0, after movie frame alignment, estimation of the microscope contrast-transfer function parameters, and manual picking particles produced reference for template picker, particles were extracted and subjected to iterative 2D classification to remove particles associated with noisy or contaminated classes. A final total of 781,859 picked particles were extracted and generated an initial model using the antibody initio reconstruction with no symmetry function in CryoSPARC. A subset of particles was selected from these 3D classes for 3D refinement.

### Model building

Model building was first conducted on the cryo-EM map of the NiV-RBP with 41-6 complex. The initial models of NiV-RBP and 41-6 were generated by SWISS-MODEL server[53] based on the PDB 7TXZ[18], and by Alphafold2-multimer[54], respectively. Then these initial models were combined and rigidly fitted into the cryo-EM map, followed by

molecular dynamics flexible fitting (MDFF)[55]. The resultant atomic model was manually optimized with Coot[56] and refined by *phenix.real_space_refine*[57]. The final model was evaluated by MolProbity[58]. Statistics of the map reconstruction and model building are shown in Supplementary Table 1.

## Reporting summary

Further information on research design is available in the Nature Portfolio Reporting Summary linked to this article.

## Data availability

The cryo-EM structures of 41-6 complexed with the NiV-RBP have been deposited in the Electron Microscopy Data Bank (EMDB) and Protein Data Bank with the accession codes EMD- 36849, PDB ID 8K3C. This study also sued 2VSM, 3D11, 6PDL, 6CMI, 7TY0, and 7TXZ from the Protein Data Bank. Other data were contained within the article/Supplementary Information. Source data are provided with this paper.

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

## Acknowledgements

We thank the staff from the Institutional Center for Shared Technologies and Facilities of Wuhan Institute of Virology, CAS for their assistance in performing flow cytometry and biolayer interferometry. We thank all the colleagues from the National Biosafety Laboratory, Wuhan, for their support during the study. We thank the Center for Biosafety Mega-Science, the Chinese Academy of Sciences, and the National Virus Resource Center for resource support. We thank the Cryo-EM Center at the University of Science and Technology of China for the support of cryo-EM data collection. This work was supported by the Strategic Priority Research Program of the Chinese Academy of Sciences (Grant No. XDB0490000, S.C., R.G., K.Z., and E.L.), the National Key Research and Development Program of China (Grant No. 2022YFC2303300, R.G.), the Key Biosafety Science and Technology Program of Hubei Jiangxia Laboratory (Grant No. JXBS002, R.G.), the Creative Research Group Program of Natural Science Foundation of Hubei Province, China (Grant No. 2022CFA021, R.G.), and the Hubei Natural Science Foundation for Distinguished Young Scholars (Grant No. 2019CFA076, R.G.).

## Author contributions

S.C. and R.G. initiated and directed the project. L.C. and Z.Z. performed phage display library screening. L.C. generated proteins, expressed antibodies, and conducted BLI assays. L.C. and Y.Q. performed binding assay. L.C., Z.C. and W.G. produced pseudovirus and carried out pseudovirus neutralization assays; H.Z., K.L. and H.Z. performed the authentic virus neutralization assays. X.Z., H.Z. and Y.Y. performed hamster challenge studies; P.F. generated the NiV tetramer RBP; M.S., M.L., S.L., E.L. and K.Z. performed structural studies on RBP/Fab complexes; L.C., M.S. and R.G. wrote the draft manuscript. S.C., R.G., K.Z. and C.Y. revised the manuscript.

## Competing interests

The remaining authors declare no competing interests.
R.G. and his colleagues are listed as inventors on pending patent applications for the identified neutralizing antibodies in this work (CN202310743583.6 and CN202310743574.7). The remaining authors declare no competing interests.
