## [Peer Review File · Nature Communications]

REVIEWER COMMENTS

Reviewer #1 (Remarks to the Author):

This manuscript reports the identification of 2 nAbs from a phage-display Fab library that with activity against NiV G and, in the case of NiV41, HeV G. Emerging henipaviruses pose a significant health threat and the development of treatment strategies including of additional biologics is of high priority. The study covers a lot of ground, ranging from discovery and biochemical characterization to epitope mapping, in vivo efficacy testing, and, for NiV41, a structural analysis of the Ab-antigen complex. However, as specified in the following, not all conclusions are supported by the experimental data, statistical analyses are missing or incomplete, rigor is lacking in the design of several experiments in which only 2 biological repeats are tested, and major concerns compromise the BLI datasets. These flaws detract substantially from the impact of the study.

Specific points:

- 1) Fig. 1a,c,d; Show replicate results and measure of experimental error in the figure. This comment equally applies to Fig. 2a,b (show individual replicates), Fig. 4 b,c (show individual replicates), and Fig. 5 a,b (show individual replicates in weight change graphs)
- 2) Fig. 1b; This BLI experiment raises multiple questions: overlapping curves at intermediate dilutions (in the top left and right graphs) suggest that saturation has been reached at these analyte concentrations. How can substantially higher signal intensities (beyond saturation) be obtained when higher analyte concentrations are tested? What does the figure legend refer to with the statement “Fab fragments were then injected over the chip surface”? This description appears to be a mixture of BLI and SPR. OctetRed dips probes into analyte dilutions; there is no fluidics and no chip in BLI. NiV42 binding curves to HeV-RBD (lower right) are non-interpretable and do not support model building. What is the value of showing binding constants extracted from these data?
- 3) Legend to Fig. 2a; “Data are represented as the mean values of duplicates and error bars represent the mean +/- S.D. Representative data are shown from two independent experiments.” Mathematically impossible to determine SD based on only two biological (independent) repeats.
- 4) Fig. 2a; Bottom row, color coding of legend and curves does not match for the left 2 graphs.
- 5) Fig. 2b; Present regression model results (IC50 concentrations) with 95% confidence intervals.
- 6) Fig. 2c; Is the graph shown in the lower right (42-27 against HeV-RBD) a copy-paste error? Graph looks identical to that shown in Fig. 1b (lower right), but claimed rate constants are different.
- 7) Fig. 2c; In addition to comment #6, superimposing the NiV-RBP vs NiV42 graph from Fig. 1b onto the “42-27 NiV-RBP” graph in Fig. 2c reveals that the shape of the four lower curves in Fig. 2c is virtually identical to that of Fig. 1b. Please provide source data for these experiments.
- 8) Fig. 3a; The statement “In general, after maturation, the mature antibodies had more resistance to alanine mutations” is not supported by the data shown in this figure.

9) Fig. 4; “Data are represented as the mean values of duplicates and error bars represent the mean +/- S.D.”. Meaningful SD cannot be calculated from n=2.

10) Clarify in figure legends that the vehicle-treated animals in Fig. 5b and Extended Data Fig. 4b,c are identical.

11) Synergy is testable. Please back up your claims experimentally or drop the statement (“Anyway, these two types of antibodies theoretically can work synergistically to prevent HNV infection”).

12) In Discussion, please further elaborate on clinical potential, taking into account your result that postexposure prophylaxis, initiated 1 dpi before the onset of clinical signs, was not statistically significantly efficacious (Extended Data Fig. 4c).

13) Provide median survival calculations for all survival figures (main and Extended Data).

Reviewer #2 (Remarks to the Author):

The manuscript by Chen et al. details the discovery and affinity maturation of two neutralizing antibodies against Nipah (NiV) and Hendra (HeV) viruses. A combination of binding assays and neutralization assays provide comprehensive data as to the potency of this line of antibody development, which is supported by live animal studies under BSL4 conditions. They further defined the epitope of one of the identified antibodies by determining a cryo-EM structure in complex with the NiV-G protein.

This is a comprehensive study, and the findings constitute a useful addition to the breadth of knowledge regarding antibody targeting of henipaviruses. The structural studies add to the currently limited amount of structural data on tetrameric NiV-G ectodomains.

The authors should address the following:

Major:

- This sentence in the results section: “Following a biolayer interferometry (BLI) binding experiment (Fig. 1b), it was ascertained that FabNiV41 is capable of binding with RBPs from both NiV and HeV, although the binding to NiV is stronger, albeit with a faster dissociation rate.” – is incorrect as written as the dissociation rate of FabNiV41 binding to NiV RBP is slower than its binding to HeV RBP. The statement should be modified to something like: “Following a biolayer interferometry (BLI) binding experiment (Fig. 1b), it was ascertained that FabNiV41 is capable of binding with RBPs from both NiV and HeV, although the binding to NiV is stronger with a slower dissociation rate.”
- For the BLI figures, the authors should show both the data and the fit of the data from which the affinity and kinetics parameters were obtained. The blank sensorgrams should also be shown, particularly for the low affinity (fast off/fast on) interactions.
- The authors state: “Simultaneously, we also conducted a comparative analysis of m102.4 (Fig. 2b).” Figure 2b does not show any data on m102.4.
- In the structural results section, it is stated that: “In our data, we found a complex of NiV-RBP with

three 41-6 Fab fragments, one above and two below (Extended Data Fig. 5b).” It is not clear from Extended Data Fig. 5b (2D class averages) that 3 Fabs are bound to the RBP. The authors should either provide clearer/ more robust evidence of 3 Fabs bound or restrict their discussion to the Fab that they do see clearly. There is only one 2D class where the 3 fabs are clearly visible, and for those not familiar with the NiV-G structure, the intermediate maps after hetero-refinement may make it seem as if the two head domains positioned proximal to the viral membrane are density from the two additional fabs. This figure could be enhanced to more clearly show where the fabs are thought to be and clarify where each head domain is in the 3D structures.

- For the cryo-EM reconstruction, the authors should provide figures showing local resolution maps. Zoomed-in views of map with underlying fitted model should be shown so an assessment can be made about the quality of the reconstruction in regions where major conclusions are being drawn.
- The authors claim that binding of 41-6 to NiV-G induces allosteric change, based on C α differences between head domains in the structure and through comparison to an apo head domain. Though the NiV-G protein is a homotetramer, it has an asymmetric structure, and given the 180° rotation in orientation of the two virion-membrane distal head domains, it may not be reasonable to assume that they would have the same conformation. The shifts noted may be a result of dimerization and not of antibody binding. Furthermore, comparison to the apo head domain (PDB 3D11) also relies on this same assumption. A superimposition with the nAH1.3-bound structure (PDB 7TXZ) would likely be more informative, and if the positioning of these dimerization residues differs between the nAH1.3 and 41-6-bound structures, it may serve as more convincing evidence of allosteric modification.

Additional minor corrections or suggestions:

- The use of HeV antigens features prominently in the study, yet HeV is only briefly mentioned in the introduction. Some discussion of the degree of similarity of HeV to NiV, both in terms of sequence and virulence, would be useful.
- The threshold was for selecting “strong binding” for the 22 chain-shifted clones should be defined.
- It appears that four antibodies were selected for efficacy against authentic virus (figure 2), although the text states five were chosen.
- In figure 2, panel a, bottom panel, it is not clear what the pink and orange curves are as they do not match any of the legends in the panel.
- It is unclear why 41-6 is being particularly noted for having improved neutralization affinity, when the EC50 values seem to be in line with the other selected NiV41 derivatives, with 41-9 having lower EC50 values against the two NiV strains.
- The abstract reports a structure of dimeric RBP with the antibody, but it seems to be that the full tetramer is bound to antibody, even if the stalk and heads from the additional two protomers are not resolved in the refined map. It would be advisable to replace the term dimer with tetramer, or at least be very clear in describing the structure of being of the dimeric head region of the full tetramer. In the discussion, it is stated that 41-6 only binds one head region, which is true when examining only the refined map used for model building, but the 2D classes and the text previously clarify that 3 fabs bind the G protein.
- In the discussion, it is stated that the one protomer with the head in the “head-up” position is responsible for receptor binding. It is not fully understood what the stoichiometry of NiV-G to Ephrin binding in a natural context is, or which protomer is first to initiate binding. This should either be rephrased or supported by a citation. Additionally, it is claimed that previous structures of receptor-binding region antibodies were of monomers, but the nAH1.3-bound structure cited is of a tetrameric G-antibody interaction.

Reviewer #3 (Remarks to the Author):

Chen et al used phage display to make a couple of new neutralizing antibodies against the Nipah virus receptor binding protein (RBP, also known as G). Such antibodies block binding between G and its receptors ephrinB2 or ephrinB3. One of these antibodies has a good level of cross-reactivity with Hendra virus, apparent using pseudotyped virus neutralization assays. The cross-reactive antibody was selected for further studies, and the authors found that the antibody could protect a certain percentage of the animals against challenge with Nipah virus, although in most cases not all the animals succumbed to the viral infection in the control untreated groups. The manuscript is of interest, but there are several major concerns, including how much better, if any at all, these antibodies are to the existing antibodies that block NiV and HeV G receptor binding. My concerns are described below:

Major:

1. From Fig. 1g, it looks like only 5 hamsters were used per group (vaccinated vs. not vaccinated). This is a very low number, particularly since only approximately 65% of the control group untreated animals succumbed to the infection. Therefore, the investigators did not use a high enough number of animals, and also the authors did not use enough challenge live virus to obtain 90% - 100% mortality. Only then the investigators could say that monoclonal antibody NiV41 “provided complete protection to the infected animals,” as said in line 139.
2. The authors said they “also conducted a comparative analysis of m102.4 (Fig. 2b). However, Fig. 2b does not have any data on antibody m102.4. Do they mean the bottom of Fig. 2a? If so, that data is neither explained nor discussed. How did the new antibodies from this study compare to m102.4? From the diagrams shown, there is no apparent improvement even after the antibodies have gone affinity maturation.
3. The statement in lines 235-238 is not accurate since not all the untreated animals (control group) succumbed to the infection: “We further investigated the efficacy of the antibody by decreasing the dosage to 3 mg/kg, and discovered that its prophylactic effect was dose dependent, leading to a survival rate of 83% among the treated hamsters,”. The % improved survival rate would be better to be noted in the text. This is also true when presenting in the main text the results of Extended Data Fig. 4.
4. It would be important to conduct a hamster survival analysis of their new mature antibody 41-6 against the best sub-clone of m104.2, to see whether or not there are any in vivo therapeutic or prophylactic improvements for the new antibody compared to the existing m104.2 antibody (see several papers by the Broder group).
5. In Fig. 6, if the structure of m104.2 bound to NiV-G is available, that should be also shown in comparison with antibody 41-6, so see the improvements of 41-6. According to Extended Data Fig. 7 there seems to show no improvement at all for 41-6 over m104.2 for competition against binding of NiV G with ephrinB2, as differences between 41-6 and m104.2 are not apparent. Only the structural comparison between 41-6 and the parental antibody of m104.2, which is m102.3, is shown in Extended Data Fig. 9. This is ok to show but not the most fair comparison, as m104.2 and 41-6 are both the more affinity mature antibodies and should be compared in both structural and functional analyses.
6. Since the binding of 41-6 appears to closely resemble ephrinB2 binding to NiV-G, does the binding of 41-6 induce the conformational changes in NiV-G that receptor binding does? This should be easily testable in collaboration with the discoverers of receptor-induced conformational changes in

NiV G.

7. Does cross-reactivity with HeV result in protection against HeV in animals?

Minor:

1. Line 72 should definitely include reference: Liu et al., 2013 (PLoS Pathogens), where the receptor-induced conformational changes in NiV G were first published.
2. The fonts in Figs. 4b and 4c are way too small to be readable.

Response to the Review Comments

Reviewer #1 (Remarks to the Author):

This manuscript reports the identification of 2 nAbs from a phage-display Fab library that with activity against NiV G and, in the case of NiV41, HeV G. Emerging henipaviruses pose a significant health threat and the development of treatment strategies including of additional biologics is of high priority. The study covers a lot of ground, ranging from discovery and biochemical characterization to epitope mapping, in vivo efficacy testing, and, for NiV41, a structural analysis of the Ab-antigen complex. However, as specified in the following, not all conclusions are supported by the experimental data, statistical analyses are missing or incomplete, rigor is lacking in the design of several experiments in which only 2 biological repeats are tested, and major concerns compromise the BLI datasets. These flaws detract substantially from the impact of the study.

Specific points:

1) Fig. 1a,c,d; Show replicate results and measure of experimental error in the figure. This comment equally applies to Fig. 2a,b (show individual replicates), Fig. 4 b,c (show individual replicates), and Fig. 5 a,b (show individual replicates in weight change graphs)

Response: We greatly appreciate your critical suggestion. We have modified the data presentation by showcasing the mean plus the standard deviation from three independent experiments. Due to the limited resource of BSL-4 laboratory, only one experiment was conducted for the animal experiments in Fig. 5a and 5b, and the data was represented by the mean (and the standard deviation) of six hamsters in each group, and this has been accepted by the field (Doyle et al., 2021; Lu et al., 2023a; Lu et al., 2023b).

Reference:

1. Doyle, M.P., Kose, N., Borisevich, V., Binshtein, E., Amaya, M., Nagel, M., Annand, E.J., Armstrong, E., Bombardi, R., Dong, J., et al. (2021). Cooperativity mediated by rationally selected combinations of human monoclonal antibodies targeting the henipavirus receptor

binding protein. Cell Rep 36, 109628.

2. Lu, M., Yao, Y., Liu, H., Zhang, X., Li, X., Liu, Y., Peng, Y., Chen, T., Sun, Y., Gao, G., et al. (2023a). Vaccines based on the fusion protein consensus sequence protect Syrian hamsters from Nipah virus infection. JCI Insight 8.
3. Lu, M., Yao, Y., Zhang, X., Liu, H., Gao, G., Peng, Y., Chen, M., Zhao, J., Zhang, X., Yin, C., et al. (2023b). Both chimpanzee adenovirus-vectored and DNA vaccines induced long-term immunity against Nipah virus infection. NPJ Vaccines 8, 170.

2) Fig. 1b; This BLI experiment raises multiple questions: overlapping curves at intermediate dilutions (in the top left and right graphs) suggest that saturation has been reached at these analyte concentrations. How can substantially higher signal intensities (beyond saturation) be obtained when higher analyte concentrations are tested? What does the figure legend refer to with the statement “Fab fragments were then injected over the chip surface”? This description appears to be a mixture of BLI and SPR. OctetRed dips probes into analyte dilutions; there is no fluidics and no chip in BLI. NiV42 binding curves to HeV-RBD (lower right) are non-interpretable and do not support model building. What is the value of showing binding constants extracted from these data?

Response: We are grateful for your suggestion. Possibly due to the limitation of the measurement, the higher signal may not be obtained even when higher concentrations are tested. Therefore, we performed the test again between several antibodies and RBPs, as shown in Fig. 1b. We have corrected the error in the figure legend (**green highlight, page 32, line 978**). The curves for the binding between NiV42 and HeV RBP are not interpretable and do not support model building ($R^2=0$), so we only present the raw data. The same applies to the binding of 42-27 and HeV RBP.

3) Legend to Fig. 2a; “Data are represented as the mean values of duplicates and error bars represent the mean +/- S.D. Representative data are shown from two independent experiments.” Mathematically impossible to determine SD based on only two biological (independent) repeats.

Response: We are appreciative of your constructive criticism. To acquire statistical analysis of the data, we conducted a number of experiments and yielded three independent replicates, displaying the results as mean+/- S.D. The description in the figure legend has also been changed accordingly.

4) Fig. 2a; Bottom row, color coding of legend and curves does not match for the left 2 graphs.

Response: We regret for our mistake and have amended the image accordingly.

5) Fig. 2b; Present regression model results (IC₅₀ concentrations) with 95% confidence intervals.

Response: We appreciate your suggestion and have included the IC₅₀ concentrations with 95% confidence intervals in Fig. 2b.

6) Fig. 2c; Is the graph shown in the lower right (42-27 against HeV-RBD) a copy-paste error? Graph looks identical to that shown in Fig. 1b (lower right), but claimed rate constants are different.

Response: We are appreciative of your constructive criticism. As demonstrated in our previous manuscript, NiV42 and 42-27 had weak binding to HeV-RBP in the BLI assay; nonetheless, when the antibodies were at high concentrations, both experiments showed similar fast-binding and fast-dissociation reactions, resulting in similar outcomes. We checked the original experimental record of BLI and found Fig. 2c and Fig. 1b were not identical but looked very similar. Actually, the calculated affinity was not accurate due to the very poor fitted curve. Hence, we reformed binding experiments between the two antibodies and HeV RBP, with the results shown in Fig. 1band 2c. The curves for the binding between NiV42 and HeV RBP are not interpretable and do not support model building ($R^2=0$), so we only present the raw data. The same applies to the binding of 42-27 and HeV RBP.

7) Fig. 2c; In addition to comment #6, superimposing the NiV-RBP vs NiV42 graph

from Fig. 1b onto the “42-27 NiV-RBP” graph in Fig. 2c reveals that the shape of the four lower curves in Fig. 2c is virtually identical to that of Fig. 1b. Please provide source data for these experiments.

Response: Thank you for the concern. The difference of the raw data between curves are very minor. So the curve looks identical. We also reperformed binding experiments between the two antibodies and NiV RBP, with the results shown in Fig. 1band 2c. The source data related to this experiment is included in the supplementary file (named “**Source Data**”).

8) Fig. 3a; The statement “In general, after maturation, the mature antibodies had more resistance to alanine mutations” is not supported by the data shown in this figure.

Response: Thanks a lot for these constructive comments. The statement has been taken away and the heading of the conclusion has been adjusted suitably (**green highlight, page 6, line 185-186**); the caption of Fig.3 has been changed too (**green highlight, page 36, line 1013**). Results from plaque reduction neutralization tests indicated that the antibodies that had undergone affinity maturation had a higher neutralization capacity than the parental antibodies, although with no marked distinction in reactivity to alanine mutations.

9) Fig. 4; “Data are represented as the mean values of duplicates and error bars represent the mean +/- S.D.”. Meaningful SD cannot be calculated from n=2.

Response: We are appreciative of your constructive criticism. To acquire statistical analysis of the data, we conducted a number of experiments and yielded three independent replicates, displaying the results as mean+/- S.D. The description in the figure legend has also been changed.

10) Clarify in figure legends that the vehicle-treated animals in Fig. 5b and Extended Data Fig. 4b,c are identical.

Response: We appreciate your suggestion and have included a description of vehicle-treated animals in the legend of Fig. 5b (**green highlight, page 40, line 1052**) and

Extended Data Fig. 4b, 4c (green highlight, page 49, line 1145).

11) Synergy is testable. Please back up your claims experimentally or drop the statement (“Anyway, these two types of antibodies theoretically can work synergistically to prevent HNV infection”).

Response: We are grateful for your suggestion and have removed the sentence accordingly.

12) In Discussion, please further elaborate on clinical potential, taking into account your result that postexposure prophylaxis, initiated 1 dpi before the onset of clinical signs, was not statistically significantly efficacious (Extended Data Fig. 4c).

Response: Thanks a lot for these constructive comments. In our study, hamsters were selected as the experimental model to evaluate the activity of antibodies. Previous research has demonstrated that hamsters display the most important characteristics of human diseases; a higher dose of challenge is needed to cause a lethal disease, leading to a quicker disease progression and a shorter time for treatment (Doyle et al., 2021). In the discussion section, we have further elaborated on the clinical potential of antibodies. “Our research has shown that 41-6 is capable of providing complete prophylactic protection to hamsters infected with NiV. Therefore, it could be used as preventive drugs during NiV pandemic. We also showed that 41-6 can be effective in the initial exposure. Then it could be used as post-exposure prophylaxis to prevent transmission since NiV is mainly spread through close contact with family and medical personnel. More animal studies are required to verify more clinical potentials of 41-6.” (green highlight, page 16, line 466-472)

Reference:

1. Doyle, M.P., Kose, N., Borisevich, V., Binshtein, E., Amaya, M., Nagel, M., Annand, E.J., Armstrong, E., Bombardi, R., Dong, J., et al. (2021). Cooperativity mediated by rationally selected combinations of human monoclonal antibodies targeting the henipavirus receptor binding protein. *Cell Rep* 36, 109628.

13) Provide median survival calculations for all survival figures (main and Extended Data).

Response: We are grateful for your suggestion and have included the median survival time of experimental animals in both the main and extended data.

Reviewer #2 (Remarks to the Author):

The manuscript by Chen et al. details the discovery and affinity maturation of two neutralizing antibodies against Nipah (NiV) and Hendra (HeV) viruses. A combination of binding assays and neutralization assays provide comprehensive data as to the potency of this line of antibody development, which is supported by live animal studies under BSL4 conditions. They further defined the epitope of one of the identified antibodies by determining a cryo-EM structure in complex with the NiV-G protein.

This is a comprehensive study, and the findings constitute a useful addition to the breadth of knowledge regarding antibody targeting of henipaviruses. The structural studies add to the currently limited amount of structural data on tetrameric NiV-G ectodomains.

The authors should address the following:

Major:

- This sentence in the results section: “Following a biolayer interferometry (BLI) binding experiment (Fig. 1b), it was ascertained that FabNiV41 is capable of binding with RBPs from both NiV and HeV, although the binding to NiV is stronger, albeit with a faster dissociation rate.” – is incorrect as written as the dissociation rate of FabNiV41 binding to NiV RBP is slower than its binding to HeV RBP. The statement should be modified to something like: “Following a biolayer interferometry (BLI) binding experiment (Fig. 1b), it was ascertained that FabNiV41 is capable of binding with RBPs from both NiV and HeV, although the binding to NiV is stronger with a slower dissociation rate.”

Response: We appreciate your suggestion and have revised the description as follows:

“Following a biolayer interferometry (BLI) binding experiment (Fig. 1b), it was ascertained that FabNiV41 is capable of binding with RBPs from both NiV and HeV,

although the binding to NiV is stronger with a slower dissociation rate.” (green highlight, page 4, line 119-121)

- For the BLI figures, the authors should show both the data and the fit of the data from which the affinity and kinetics parameters were obtained. The blank sensorgrams should also be shown, particularly for the low affinity (fast off/fast on) interactions.

Response: We are deeply thankful for your suggestion. The data for the BLI in Fig. 1b and Fig. 2c was presented with both raw and fit data. Data analysis for binding between antibodies and RBPs was conducted after subtracting the non-specific binding using reference biosensor. However, the curves for the binding between NiV42 and HeV RBP are not interpretable and do not support model building ($R^2=0$), so we only present the raw data. The same applies to the binding of 42-27 and HeV RBP.

To assess the specificity for the low affinity interactions especially for the fast-on/fast-off interactions, a reaction between RBP and high concentrations of antibodies with blank biosensors was carried out, and the results, as displayed in Extended Fig 2a, indicated no non-specific binding when NiV42 and 42-27 were exposed to HeV-RBP.

- The authors state: “Simultaneously, we also conducted a comparative analysis of m102.4 (Fig. 2b).” Figure 2b does not show any data on m102.4.

Response: Our sincere apologies for the mistake we made and we have already conveyed that the data of m102.4 has been incorporated into Fig. 2a.

- In the structural results section, it is stated that: “In our data, we found a complex of NiV-RBP with three 41-6 Fab fragments, one above and two below (Extended Data Fig. 5b).” It is not clear from Extended Data Fig. 5b (2D class averages) that 3 Fabs are bound to the RBP. The authors should either provide clearer/ more robust evidence of 3 Fabs bound or restrict their discussion to the Fab that they do see clearly. There is only one 2D class where the 3 fabs are clearly visible, and for those not familiar with the NiV-G structure, the intermediate maps after hetero-refinement may make it seem as if the two head domains positioned proximal to the viral membrane are density from

the two additional fabs. This figure could be enhanced to more clearly show where the fabs are thought to be and clarify where each head domain is in the 3D structures.

Response: We thank this reviewer for pointing this out. We have modified the Extended Data Fig. 5 to clearly illustrate the binding location of three Fabs to RBP by comparing the NiV-RBP ectodomain tetramer and the complex of NiV-RBP/41-6 (the 2D class averages in the Extended Data Fig. 5a, 5b). In addition, we add arrows indicating the Fabs and heads from the 3D structure (Extended Data Fig. 5c).

Although we observed two complexes in the data, one NiV-RBP binding to one Fab and another NiV-RBP binding to three Fabs, limited by our data, we reconstructed the NiV-RBP binding to one Fab based on the structure of virus-host physiological infection. We have revised the text “In our data, we found a complex of NiV-RBP with three 41-6 Fab fragments, one above and two below (Extended Data Fig. 5b).” to “In the complex structure, we noticed that a portion of the NiV-RBP tetramer was bound to three Fabs (Extended Data Fig. 5a, 5b). However, due to the flexibility and physiological nature of viral-host infection, we primarily analyzed the structure of NiV-RBP binding with one Fab (Fig. 6a and Extended Data Fig. 5).” (**green highlight, page 9, line 266-270**).

- For the cryo-EM reconstruction, the authors should provide figures showing local resolution maps. Zoomed-in views of map with underlying fitted model should be shown so an assessment can be made about the quality of the reconstruction in regions where major conclusions are being drawn.

Response: We have included the local resolution map (Extended Data Fig. 5d) and the zoomed-in views of map with fitted model in Fig. 6a.

- The authors claim that binding of 41-6 to NiV-G induces allosteric change, based on C α differences between head domains in the structure and through comparison to an apo head domain. Though the NiV-G protein is a homotetramer, it has an asymmetric structure, and given the 180° rotation in orientation of the two virion-membrane distal head domains, it may not be reasonable to assume that they would have the same

conformation. The shifts noted may be a result of dimerization and not of antibody binding. Furthermore, comparison to the apo head domain (PDB 3D11) also relies on this same assumption. A superimposition with the nAH1.3-bound structure (PDB 7TXZ) would likely be more informative, and if the positioning of these dimerization residues differs between the nAH1.3 and 41-6-bound structures, it may serve as more convincing evidence of allosteric modification.

Response: We thank this reviewer for the insightful suggestion. In our revised text, we have removed the conclusion to avoid any concern and added the comparison with nAH1.3-bound NiV-RBP structure in Extended Data Fig.6b as well as the revised text (green highlight, page 10, line 285-292).

Additional minor corrections or suggestions:

- The use of HeV antigens features prominently in the study, yet HeV is only briefly mentioned in the introduction. Some discussion of the degree of similarity of HeV to NiV, both in terms of sequence and virulence, would be useful.

Response: We are grateful for your input, and have included a description of HeV in the discussion accordingly: “Hendra virus, a fatally zoonotic member of the genus Henipavirus, was first identified in Australia in 1994. It has caused sporadic outbreaks in Australia, leading to seven human infections including four cases of death. due to fatal encephalitis, giving a mortality rate of 57% (Li et al., 2023). It has been determined that human infections are related to close contact with sick horses. The HeV RBP is 604 amino acids in length and has 83% similarity to NiV (Wang et al., 2001).” (green highlight, page 12, line 358-363).

- The threshold was for selecting “strong binding” for the 22 chain-shifted clones should be defined.

Response: In this study, strong binding was defined as having an EC₅₀ of 0.5 µg/ml or less as determined by ELISA experiments. Therefore, clones with an EC₅₀ below this threshold were selected. We have added this definition in the results section. (green highlight, page 5, line 148-149)

- It appears that four antibodies were selected for efficacy against authentic virus (figure 2), although the text states five were chosen.

Response: We regret the mistake and have updated the main text accordingly. (green highlight, page 6, line 162-163)

- In figure 2, panel a, bottom panel, it is not clear what the pink and orange curves are as they do not match any of the legends in the panel.

Response: We regret our mistake and have amended the image accordingly.

- It is unclear why 41-6 is being particularly noted for having improved neutralization affinity, when the EC50 values seem to be in line with the other selected NiV41 derivatives, with 41-9 having lower EC50 values against the two NiV strains.

Response: Thanks a lot for the critical suggestions. Our study revealed that, despite no considerable variation in the binding and neutralizing activity of the two antibodies, 41-6 and 41-9, higher expression level of 41-6 was observed under the same expression conditions. Consequently, we decided to use 41-6 for subsequent animal experiments and structural experiments. To avoid any ambiguity, we have removed the particular instructions for 41-6 from the text. (page 6, line 165)

- The abstract reports a structure of dimeric RBP with the antibody, but it seems to be that the full tetramer is bound to antibody, even if the stalk and heads from the additional two protomers are not resolved in the refined map. It would be advisable to replace the term dimer with tetramer, or at least be very clear in describing the structure of being of the dimeric head region of the full tetramer. In the discussion, it is stated that 41-6 only binds one head region, which is true when examining only the refined map used for model building, but the 2D classes and the text previously clarify that 3 fabs bind the G protein.

Response: We are grateful for the suggestion. We have made revisions to the abstract accordingly. The new sentence is that “Furthermore, a 2.88 Å Cryo-EM structure of the

tetrameric RBP and antibody complex demonstrated that 41-6 blocked the receptor binding interface.” (green highlight, page 2, line 47-49).

As mentioned in the structural results section, we observed two complexes in the data: NiV-RBP binding to one Fab or to three Fabs. Due to limitations in our data, we reconstructed the NiV-RBP binding to one Fab based on the structure of virus-host physiological infection. This information has been incorporated into the discussion to provide a contextual understanding. Now we have modified to “Our structural data revealed that the soluble NiV-RBP ectodomain has two heads near the viral terminal and two heads near the host terminal, which bind with either one or three 41-6 Fabs. Given resolution limitations and the structural basis of physiological viral-host infection, we resolved a Cryo-EM of single antibody (41-6 Fab)-bound homotetramer NiV-RBP, which showed that the two head regions distant from the viral membrane form an asymmetric dimer.” (green highlight, page 14, line 408-414).

- In the discussion, it is stated that the one protomer with the head in the “head-up” position is responsible for receptor binding. It is not fully understood what the stoichiometry of NiV-G to Ephrin binding in a natural context is, or which protomer is first to initiate binding. This should either be rephrased or supported by a citation. Additionally, it is claimed that previous structures of receptor-binding region antibodies were of monomers, but the nAH1.3-bound structure cited is of a tetrameric G-antibody interaction.

Response: We understand the reviewer’s concern. In our revised text, we have replaced the sentence with “It was observed that one of the four head domains directs its receptor binding site towards the host cell membrane, while the remaining three sites are oriented towards the viral membrane.” (green highlight, page 14, line 414-416).

For the second question from original text, we want to try revising the text to clarify the question avoiding any concern. Firstly, since what we are concerned about is analyzing the occupied epitope information on the RBP head domain, we compared the structures of NiV-RBP bound with receptors or antibodies. It is worth noting that the previously reported structures with the receptor or the antibodies are from RBP head

domains which are not tetramers, except for nAH1.3. In our structure, 41-6 is the other antibody bound to NiV-RBP tetramer.

When comparing the epitopes on the head domain, we found that the epitope occupied by 41-6 on the head overlaps with the receptor or receptor-binding antibody. Therefore, 41-6 antibody acts as a receptor-competitive antibody that blocks receptor binding to neutralize the virus. Meanwhile the change in overall head conformation is not very significant, but slightly different in some regions. Besides we want to know whether and how 41-6 as a receptor-competitive antibody inducing the rearrangement, it seems by depending on the shift of stalk and neck domain compared with complex of NiV-RBP tetramer bound with nAH1.3 (seen in Extended Fig.9e).

We have revised the text as “This type of binding has not been documented previously, and upon further investigation. We found the epitope of 41-6 is similar to the binding sites in RBP of the receptor and other receptor-like antibodies. It was also disclosed that the binding of 41-6 primarily induces a reconfiguration of the stalk and neck domains.” (green highlight, page 14, line 420-424)

Reviewer #3 (Remarks to the Author):

Chen et al used phage displays to make a couple of new neutralizing antibodies against the Nipah virus receptor binding protein (RBP, also known as G). Such antibodies block binding between G and its receptors ephrinB2 or ephrinB3. One of these antibodies has a good level of cross-reactivity with Hendra virus, apparent using pseudotyped virus neutralization assays. The cross-reactive antibody was selected for further studies, and the authors found that the antibody could protect a certain percentage of the animals against challenge with Nipah virus, although in most cases not all the animals succumbed to the viral infection in the control untreated groups. The manuscript is of interest, but there are several major concerns, including how much better, if any at all, these antibodies are to the existing antibodies that block NiV and HeV G receptor binding. My concerns are described below:

Major:

1. From Fig. 1g, it looks like only 5 hamsters were used per group (vaccinated vs. not

vaccinated). This is a very low number, particularly since only approximately 65% of the control group untreated animals succumbed to the infection. Therefore, the investigators did not use a high enough number of animals, and also the authors did not use enough challenge live virus to obtain 90% - 100% mortality. Only then the investigators could say that monoclonal antibody NiV41 “provided complete protection to the infected animals,” as said in line 139.

Response: In Fig. 1g, we used six hamsters per group and 5-6 animals per group have been widely accepted in the field (Doyle et al., 2021; Lu et al., 2023a; Lu et al., 2023b). According to earlier studies, the LD₅₀ of hamsters by IP route was 270 pfu (Wong et al., 2003). In our study, we increased the virus to 10⁵ TCID₅₀ ($\approx 7 \times 10^4$ pfu), yet the mortality rate did not exceed 90%. This could be due to the various virus strains and animals used in different laboratories, as reported (Mire et al., 2016). As this experiment was only a preliminary test to validate antibody activity, we did not increase the virus dose to evaluate the protective activity of NiV41. Accordingly, we have adjusted the description “Our results indicate that administration at a dose of 3 mg/kg six hours post virus exposure provided complete protection to the infected animals” to “Our results indicate that administration at a dose of 3 mg/kg six hours post virus exposure provided significant protection to the infected animals.” (**green highlight, page 5, line 140-142**)

Reference:

1. Doyle, M.P., Kose, N., Borisevich, V., Binshtein, E., Amaya, M., Nagel, M., Annand, E.J., Armstrong, E., Bombardi, R., Dong, J., et al. (2021). Cooperativity mediated by rationally selected combinations of human monoclonal antibodies targeting the henipavirus receptor binding protein. *Cell Rep* 36, 109628.
2. Lu, M., Yao, Y., Liu, H., Zhang, X., Li, X., Liu, Y., Peng, Y., Chen, T., Sun, Y., Gao, G., et al. (2023a). Vaccines based on the fusion protein consensus sequence protect Syrian hamsters from Nipah virus infection. *JCI Insight* 8.
3. Lu, M., Yao, Y., Zhang, X., Liu, H., Gao, G., Peng, Y., Chen, M., Zhao, J., Zhang, X., Yin, C., et al. (2023b). Both chimpanzee adenovirus-vectored and DNA vaccines induced long-term immunity against Nipah virus infection. *NPJ Vaccines* 8, 170.

4. Wong, K.T., Grosjean, I., Brisson, C., Blanquier, B., Fevre-Montange, M., Bernard, A., Loth, P., Georges-Courbot, M.C., Chevallier, M., Akaoka, H., et al. (2003). A golden hamster model for human acute Nipah virus infection. *Am J Pathol* 163, 2127-2137.
5. Mire, C.E., Satterfield, B.A., Geisbert, J.B., Agans, K.N., Borisevich, V., Yan, L., Chan, Y.P., Cross, R.W., Fenton, K.A., Broder, C.C., et al. (2016). Pathogenic Differences between Nipah Virus Bangladesh and Malaysia Strains in Primates: Implications for Antibody Therapy. *Sci Rep* 6, 30916.

2. The authors said they “also conducted a comparative analysis of m102.4 (Fig. 2b). However, Fig. 2b does not have any data on antibody m102.4. Do they mean the bottom of Fig. 2a? If so, that data is neither explained nor discussed. How did the new antibodies from this study compare to m102.4? From the diagrams shown, there is no apparent improvement even after the antibodies have gone affinity maturation.

Response: Thanks a lot for the concerns and questions. The results of the neutralization test of m102.4 are depicted in Figure 2a and the IC_{50} values are discussed in detail in the text (green highlight, page 6, line 168-169). Results from BLI and plaque reduction neutralization experiments showed that the antibodies, 41-4, 41-6 and 41-9 that acquired through affinity maturation had a greater affinity and neutralization capacity than the maternal antibody NiV41. Despite the lack of a discernible increase in affinity for 42-27 compared to its maternal antibody NiV42, its neutralization activity has enhanced. Results showed that m102.4 had a more effective neutralizing capacity towards NiV than HeV (NiV_B, 0.025 μ g/ml; NiV_M, 0.013 μ g/ml; HeV, 0.559 μ g/ml). Through the light chain shuffling process, matured antibody 41-6 displayed impressive cross neutralization against Henipavirus, with equivalent neutralization against NiV and HeV (NiV_B, 0.17 μ g/ml; NiV_M, 0.08 μ g/ml; HeV, 0.22 μ g/ml). Comparatively, matured antibody 42-27 had a similar neutralization activity to m102.4, yet had a lower IC_{90} value when neutralizing NiV_M (IC_{90} : 42-27, 0.19 μ g/ml; m102.4, 2.46 μ g/ml). The manuscript has been expanded with further elucidation.

“The NiV41-derived antibodies 41-4, 41-6, and 41-9 are a class of broad-spectrum antibodies against HeV and NiV that are comparable to m102.4, while 42-27 is

particularly effective in neutralizing NiV. Results from the NiV41-derived antibodies showed an equivalent level of neutralizing activity against both HeV and NiV, with the capacity to neutralize more than 90% of viruses. Nevertheless, m102.4 was found to have a better effect against NiV. However, it is difficult for m102.4 to reach a 90% neutralizing effect against HeV at the IC₉₀ concentration of NiV41-derived antibodies.”
(green highlight, page 6, line 169-177)

“Results revealed that NiV41-derived antibodies showed an increase in affinity for RBPs, whereas 42-27 maintained its affinity with the maternal antibody but had a greater neutralizing activity.” **(green highlight, page 6, line 180-183)**

3. The statement in lines 235-238 is not accurate since not all the untreated animals (control group) succumbed to the infection: “We further investigated the efficacy of the antibody by decreasing the dosage to 3 mg/kg, and discovered that its prophylactic effect was dose dependent, leading to a survival rate of 83% among the treated hamsters.”. The % improved survival rate would be better to be noted in the text. This is also true when presenting in the main text the results of Extended Data Fig. 4.

Response: We are appreciative of your idea and have adjusted the survival rate description of treated animals accordingly: “We further investigated the efficacy of the antibody by decreasing the dosage to 3 mg/kg, and discovered that its prophylactic effect was dose dependent, leading to 67% improved survival rate among the treated hamsters” **(green highlight, page 9, line 247-250)**

“In this severe infection model, the survival rate of the animals was improved 50% when 10 mg/kg of antibody was administered intraperitoneally three hours after infection (Extended Data Fig. 4b). When the treatment was applied at three hours and three days postinfection, the survival rate of the animals was further improved by 67% (Fig. 5b).” **(green highlight, page 9, line 252-257)**

4. It would be important to conduct a hamster survival analysis of their new mature antibody 41-6 against the best sub-clone of m104.2, to see whether or not there are any in vivo therapeutic or prophylactic improvements for the new antibody compared to the

existing m104.2 antibody (see several papers by the Broder group).

Response: Thanks a lot for the constructive comments. Owing to the busy schedule of the BSL-4 laboratory, currently we were only able to evaluate the neutralization activity of 41-6 and m102.4 through cell-based neutralization experiments. Unfortunately, we are unable to schedule another animal experiment in the near future due to the limited resource. We discovered that 41-6 could neutralize up to 90% of HeV, whereas m102.4 was not able to achieve the same level of efficacy at a concentration of 5 µg/ml. In the future study, we will continue to evaluate and compare the protective effects of 41-6 and m102.4 in the hamster model against NiV and HeV. Additionally, we have discussed the efficiencies of 41-6 and m102.4 in the discussion section as follows:

“For NiV receptor-like antibodies, m102.4 has been shown to be protective in ferrets and AGMs, while HENV26 has been demonstrated to be efficacious in ferrets, and 41-6 has been found to be protective in hamsters. To gain a better understanding of the differences in their efficacy in vivo, a more thorough comparison under the same experimental conditions is necessary.” (green highlight, page 16, line 472-477)

5. In Fig. 6, if the structure of m104.2 bound to NiV-G is available, that should be also shown in comparison with antibody 41-6, so see the improvements of 41-6. According to Extended Data Fig. 7 there seems to show no improvement at all for 41-6 over m104.2 for competition against binding of NiV G with ephrinB2, as differences between 41-6 and m104.2 are not apparent. Only the structural comparison between 41-6 and the parental antibody of m104.2, which is m102.3, is shown in Extended Data Fig. 9. This is ok to show but not the most fair comparison, as m104.2 and 41-6 are both the more affinity mature antibodies and should be compared in both structural and functional analyses.

Response: We thank this reviewer for the suggestion. Unfortunately, we don't have the structure of m102.4 bound to NiV-RBP (Extended Data Fig. 9). Since the amino acid sequence of m102.3 and m102.4 is very similar (**Figure 1**), with the identical heavy chain and the slightly different light chain, the structure of predicted m102.4 bound to NiV-RBP is quite similar with m102.3 bound to NiV-G (**Figure 2**) (Tit-Oon et al., 2020).

So, we use the structure of m102.3 bound to NiV-RBP for the comparison in Extended Fig. 9. To provide a clearer understanding, we have also included the 102.3 structure for comparison in the main text. “m102.4 is a matured antibody that shares the same parental antibody clone with m102.3. Computer simulations indicated that their structure is very similar. After comparison of the structure of the complexes of m102.3/RBP and 41-6/RBP, we can conclude 41-6 is also a potential therapeutic antibody like m102.4 targeting receptor epitopes.” (green highlight, page 12, line 338-342)

	1	10	20	30	40	50	60
m102.3-heavy	EVQLVQSGAEVKKRQSSVKVCSKSSGGTFSNYAINWVRQAPGQGLEWMGGIIFILGIANY						
m102.4-heavy	EVQLVQSGAEVKKRQSSVKVCSKSSGGTFSNYAINWVRQAPGQGLEWMGGIIFILGIANY						
	70	80	90	100	110	120	
m102.3-heavy	AQLFQGRVITITDESTSTAYMELSSLRSEDAVYYCARQNGRQLAPHPSSQYYYYYGMG						
m102.4-heavy	AQLFQGRVITITDESTSTAYMELSSLRSEDAVYYCARQNGRQLAPHPSSQYYYYYGMG						
	130	140	150	160	170	180	
m102.3-heavy	VWGQGTITVTVSSASTKGFVFPFLAPSSKSTSGGTAALGCLVKDYFPEPPTVSWNSGALTS						
m102.4-heavy	VWGQGTITVTVSSASTKGFVFPFLAPSSKSTSGGTAALGCLVKDYFPEPPTVSWNSGALTS						
	190	200	210	220	230		
m102.3-heavy	GVHTFPAVLQSSGLYSLSVTVFPSSSLGTQTYICNVNHRPSTKVDKKEVPEPKSC						
m102.4-heavy	GVHTFPAVLQSSGLYSLSVTVFPSSSLGTQTYICNVNHRPSTKVDKKEV.....						
	1	10	20	30	40	50	60
m102.3-light	EIVMTQSPGSLSPGERATLSCRASQSHRSYVLAWYQQRPGQAPRLIYCASSRATGIP						
m102.4-light	EIVMTQSPGSLSPGERATLSCRASQSRNNYVLAWYQQRPGQAPRLIYNGSIRATGIP						
	70	80	90	100	110		
m102.3-light	DRFSGSGSGTDFTLTISRLEPEDFAVYYCQQYGRMPELSSFGCSTKVEIKRTVAAPSVFIF						
m102.4-light	DRFSGSGSGTDFTLTISRLEPEDFAVYYCQQYGMRRVHSGCSTKVEIKRTVAAPSVFIF						
	120	130	140	150	160	170	
m102.3-light	PPSDRQLKSGTASVVCLLNNFPREAKVQWYVDNALQSGNSQESVTEQDSKDSTYSLSST						
m102.4-light	PPSDRQLKSGTASVVCLLNNFPREAKVQWYVDNALQSGNSQESVTEQDSKDSTYSLSST						
	180	190	200	210			
m102.3-light	LTLTKADYEKHKLYACEVTHQGLSSPVTKSFNRGEC						
m102.4-light	LTLTKADYEKHKLYACEVTHQGLSSPVTKSFNRGEC						

Figure 1. Amino acid sequences alignment between m102.3 and m102.4.

Figure 2. Computational docking and mode of m102.4-RBP binding. (Tit-Oon et al., 2020)

Reference:

1. Tit-Oon, P., Tharakaraman, K., Artpradit, C., Godavarthi, A., Sungkeeree, P., Sasisekharan, V., Kerdwong, J., Miller, N.L., Mahajan, B., Khongmanee, A., et al. (2020). Prediction of the binding interface between monoclonal antibody m102.4 and Nipah attachment glycoprotein using structure-guided alanine scanning and computational docking. *Scientific Reports* 10, 18256.

6. Since the binding of 41-6 appears to closely resemble ephrinB2 binding to NiV-G, does the binding of 41-6 induce the conformational changes in NiV-G that receptor binding does? This should be easily testable in collaboration with the discoverers of receptor-induced conformational changes in NiV G.

Response: We gratefully appreciate the reviewer's comment. To answer the first question, we base our conclusion on the structural comparison of receptor and antibody binding to the NiV-RBP head domain (Extended Data Fig. 8a, b, c, d). We find that their binding patterns are similar. For the NiV-RBP ectodomain tetramer, the tetramer comprises four protomers, one of which includes the neck domain, linker stalk domain, and head domain. Any change to any of these parts, not just the head domain, would induce a change in overall conformation. When the NiV-RBP head domain binds with 41-6 or without, there are some regions that deviate, which may be caused by dimerization itself or by 41-6-induced (Extended Data Fig. 8c). Additionally, When the NiV-RBP head domain binds with nAH1.3 or 41-6, there are some shifts in the head, stalk, and neck domains (Extended Data Fig. 9e), which changes are induced by 41-6. Although 41-6 generates conformational changes, it is still not clear what type of changes NiV-RBP undergoes when bound with 41-6 before or after in insight to structure.

Evidence for receptor-induced NiV RBP rearrangement comes from changes in RBP circular dichroism, Raman spectra, second harmonic generation, and binding affinity to a panel of conformation-specific anti-RBP antibodies (Wong et al., 2017). Given the

restrictions of experimental materials and instruments, we opted for circular dichroism to detect the conformational alterations resulting from secondary structural changes in RBP protein. Our findings indicate that the monomer RBP head domain was not affected by ephrinB2 binding, however the tetramer extracellular RBP underwent changes, which is consistent with prior research (Aguilar et al., 2009). Moreover, 41-6 is able to cause minor monomeric alterations and can lead to changes in the tetramers. To gain further insight, we aim to explore how and what 41-6 triggers the conformational change, similarly to ephrinB2, through obtaining the structural information of NiV-RBP tetramer binding Fabs or of ephrinB2 in future studies.

Figure 3. Changes in the secondary structure of RBP monomer or tetramer were monitored using circular dichroism upon the addition of ephrinB2 or 41-6 Fab fragment. A theoretical spectrum is the sum of RBP and 41-6 (or ephrinB2) scans. The difference between the latter two lines reflects differences in secondary structure.

Reference:

1. Aguilar, H.C., Ataman, Z.A., Aspericueta, V., Fang, A.Q., Stroud, M., Negrete, O.A., Kammerer, R.A., and Lee, B. (2009). A Novel Receptor-induced Activation Site in the Nipah Virus Attachment Glycoprotein (G) Involved in Triggering the Fusion Glycoprotein (F). *Journal of Biological Chemistry* 284, 1628-1635.
2. Wong, J.J.W., Young, T.A., Zhang, J., Liu, S., Leser, G.P., Komives, E.A., Lamb, R.A., Zhou, Z.H., Salafsky, J., and Jardetzky, T.S. (2017). Monomeric ephrinB2 binding induces allosteric changes in Nipah virus G that precede its full activation. *Nat Commun* 8, 781.

7. Does cross-reactivity with HeV result in protection against HeV in animals?

Response: We express our appreciation for the reviewer's comment. Due to the busy schedule of the BSL-4 laboratory, we are unable to determine whether 41-6 has any cross-protective activity against HeV in animals in the near future due to the limited resource. In the future study, we will continue to evaluate the protective effects of 41-6

in the hamster model against HeV.

Minor:

1. Line 72 should definitely include reference: Liu et al., 2013 (PLoS Pathogens), where the receptor-induced conformational changes in NiV G were first published.

Response: Thanks a lot for the critical suggestions. We have included the reference to demonstrate the receptor-induced conformations changes in NiV RBP. (green highlight, page 3, line 72)

2. The fonts in Figs. 4b and 4c are way too small to be readable.

Response: Thanks a lot for the constructive comments. The figure has been revised for improved readability.

REVIEWERS' COMMENTS

Reviewer #1 (Remarks to the Author):

The manuscript is improved, but several questions remain:

- 1) It is appreciated that ABSL-4 containment poses constraints on animal numbers. Six hamsters/group constitutes an acceptable sample size under these conditions. However, please show measurements for all individual animals in figures 5a and 5b (symbols equal individual animals), rather than symbols equal means, which has been accepted as best practice for very small sample sizes such as yours.
- 2) The original legend to figure 4b,c stated that shown are "mean values of duplicates and error bars represent the mean \pm S.D.". The revised legend now claims that "Data are represented as the mean \pm S.D from n=3 independent experiments.". In your rebuttal letter, you declared that "To acquire statistical analysis of the data, we conducted a number of experiments and yielded three independent replicates, displaying the results as mean \pm - S.D. The description in the figure legend has also been changed." Comparing the graphs of the original figure 4b and 4c with the revised version, there is absolutely zero difference in position of symbols, shapes of curves, or size of error bars in any of the 12 graphs shown. I certainly appreciate the reproducibility of your experiments, but am still left wondering whether that is even possible?
- 3) Are the differences in median survival between animals in the PBS and 41-6 groups shown in Extended Data Fig 4b and 4c statistically significant? Please add P values also to these graphs. If not statistically significant, your description of Extended Data Fig 4b ("the survival rate of the animals was improved 50% when 10 mg/kg of antibody was administered intraperitoneally three hours after infection (Extended Data Fig. 4b)") is misleading. How do you know that anything was meaningfully "improved" should the effect lack significance? Please consider this problem also in the Discussion section of possible clinical sue of 46-1.
- 4) In your response to previous comment #12, you state that "Then it could be used as post-exposure prophylaxis to prevent transmission [...]". Was suppression of virus transmission by 46-1 experimentally confirmed? Is so, please show the data. If not, please revise this section.

Reviewer #2 (Remarks to the Author):

The authors have made efforts to address reviewer critiques in this revised manuscript. A few points still remain that should be addressed to clarify the figures and text further:

1. Extended Figure 2A legend needs more details and clarity.
2. IC50 value for m102.4 should be listed in Figure 2B.
3. Lines 266-270 needs to be re-written for accuracy and clarity. Suggest rewording as follows: "In the complex structure, we noticed that the NiV-RBP tetramer was bound to three Fabs (Extended Data Fig. 5a, 5b). However, due to the flexibility and physiological nature of viral-host infection, to obtain better resolution we primarily analyzed the structure of one of the bound Fabs (Fig. 6a and Extended Data Fig. 5)."
4. Lines 288-290: "The overall head structures were similar, although there were some regions that deviated, possibly due to the dimerization and allosteric effects induced by 41-6." There is no evidence to support that 41-6 binding causes head dimerization. This claim should either be further supported or removed.
5. Lines 358-363 are better placed in the Introduction.
6. Lines 408-410: recommend rephrasing as follows: "Our structural data revealed two populations where the soluble NiV-RBP ectodomain that has two heads near the viral terminal and two heads near the host terminal, was bound with either one or three 41-6 Fabs."
7. Lines 417-420: "In contrast to non-receptor-competitive antibody nAH1.3, our 41-6, a receptor-competitive antibody, is limited to binding to only one head region, possibly because the binding epitope in the other protomer is obstructed by the adjacent subunit." This is incorrect. Although the high-resolution structure is of one Fab, the data (Extended Figure 5, panel b) show up to 3 41-

6 bound to NiV-RBP.

8. Lines 420-423: "This type of binding has not been documented previously, and upon further investigation. We found the epitope of 41-6 is similar to the binding sites in RBP of the receptor and other receptor-like antibodies." This is a fragmented sentence and should be revised. References must be provided for the other structures that are being compared. A figure must be shown for this comparison.

Reviewer #3 (Remarks to the Author):

My prior concerns were addressed in an ok manner.

It would have been nice to perform the additional BSL-4 experiments suggested, but I understand the time it will take.

The fonts in several of the figures are still too tiny and hard to read.

Response to the Review Comments

Reviewer #1 (Remarks to the Author):

The manuscript is improved, but several questions remain:

1) It is appreciated that ABSL-4 containment poses constraints on animal numbers. Six hamsters/group constitutes an acceptable sample size under these conditions. However, please show measurements for all individual animals in figures 5a and 5b (symbols equal individual animals), rather than symbols equal means, which has been accepted as best practice for very small sample sizes such as yours.

Response: We are grateful for your suggestion. Upon thorough review of relevant literature (Bangaru et al., 2019; Lee et al., 2016; Li et al., 2022; Sun et al., 2022; Moin et al., 2022), it has been observed that experiments involving 5 or 6 animals typically present their weight change data in the form of mean \pm S.D. Consequently, in order to align with the statistical methodology commonly used in such cases, we have chosen to maintain the presentation of the data in mean \pm S.D format.

Reference

1. Bangaru, S., Lang, S., Schotsaert, M., Vanderven, H.A., Zhu, X., Kose, N., Bombardi, R., Finn, J.A., Kent, S.J., Gilchuk, P., et al. (2019). A Site of Vulnerability on the Influenza Virus Hemagglutinin Head Domain Trimer Interface. *Cell* 177, 1136-1152 e1118.
2. Lee, J., Boutz, D.R., Chromikova, V., Joyce, M.G., Vollmers, C., Leung, K., Horton, A.P., DeKosky, B.J., Lee, C.H., Lavinder, J.J., et al. (2016). Molecular-level analysis of the serum antibody repertoire in young adults before and after seasonal influenza vaccination. *Nat Med* 22, 1456-1464.
3. Li, T., Chen, J., Zheng, Q., Xue, W., Zhang, L., Rong, R., Zhang, S., Wang, Q., Hong, M., Zhang, Y., et al. (2022). Identification of a cross-neutralizing antibody that targets the receptor binding site of H1N1 and H5N1 influenza viruses. *Nat Commun* 13, 5182.
4. Sun, X., Liu, C., Lu, X., Ling, Z., Yi, C., Zhang, Z., Li, Z., Jin, M., Wang, W., Tang, S., et al. (2022). Unique binding pattern for a lineage of human antibodies with broad reactivity against influenza A virus. *Nat Commun* 13, 2378.
5. Moin SM, Boyington JC, Boyoglu-Barnum S, et al. (2022) Co-immunization with

hemagglutinin stem immunogens elicits cross-group neutralizing antibodies and broad protection against influenza A viruses. *Immunity* 55, 2405-2418

2) The original legend to figure 4b,c stated that shown are “mean values of duplicates and error bars represent the mean \pm S.D.”. The revised legend now claims that “Data are represented as the mean \pm S.D from n=3 independent experiments.”. In your rebuttal letter, you declared that “To acquire statistical analysis of the data, we conducted a number of experiments and yielded three independent replicates, displaying the results as mean \pm - S.D. The description in the figure legend has also been changed.” Comparing the graphs of the original figure 4b and 4c with the revised version, there is absolutely zero difference in position of symbols, shapes of curves, or size of error bars in any of the 12 graphs shown. I certainly appreciate the reproducibility of your experiments, but am still left wondering whether that is even possible?

Response: We regret for our mistake and have made correction to the data presented in Figure 4.

3) Are the differences in median survival between animals in the PBS and 41-6 groups shown in Extended Data Fig 4b and 4c statistically significant? Please add P values also to these graphs. If not statistically significant, your description of Extended Data Fig 4b (“the survival rate of the animals was improved 50% when 10 mg/kg of antibody was administered intraperitoneally three hours after infection (Extended Data Fig. 4b)”) is misleading. How do you know that anything was meaningfully “improved” should the effect lack significance? Please consider this problem also in the Discussion section of possible clinical use of 46-1.

Response: Thanks a lot for the critical suggestions. The P values for Extended Figure 4 have been included. Since there was no statistically significant difference in median survival between animals in the PBS and 41-6 groups as shown in extended figure 4b, the description of this result has been adjusted as “In this model of severe infection, administering 10 mg/kg of antibody intraperitoneally three hours after infection resulted in four out of six hamsters surviving (Supplementary Fig. 4b). When the

treatment was given twice at three hours and three days post-infection, five of the six hamsters remained alive.” (green highlight, page 9, line 258-262).

4) In your response to previous comment #12, you state that “Then it could be used as post-exposure prophylaxis to prevent transmission [...]”. Was suppression of virus transmission by 46-1 experimentally confirmed? Is so, please show the data. If not, please revise this section.

Response: Thanks a lot for the critical suggestions. We agree with the reviewer’s comments and re-wrote this sentence as “Then it could be used as a post-exposure prophylactic treatment for high-risk populations” (green highlight, page 16, line 466-468)

Reviewer #2 (Remarks to the Author):

The authors have made efforts to address reviewer critiques in this revised manuscript. A few points still remain that should be addressed to clarify the figures and text further:

1. Extended Figure 2A legend needs more details and clarity.

Response: We appreciate your suggestion and have added more details and clarity in legend of Extended Figure 2a (renamed Supplementary Figure 2a).

2. IC₅₀ value for m102.4 should be listed in Figure 2B.

Response: Thanks a lot for these constructive comments. We have added the IC₅₀ values of m102.4 in figure 2b.

3. Lines 266-270 needs to be re-written for accuracy and clarity. Suggest rewording as follows: “In the complex structure, we noticed that the NiV-RBP tetramer was bound to three Fabs (Extended Data Fig. 5a, 5b). However, due to the flexibility and physiological nature of viral-host infection, to obtain better resolution we primarily analyzed the structure of one of the bound Fabs (Fig. 6a and Extended Data Fig. 5).”

Response: We appreciate your suggestion and have made the modification to this description as per your suggestion. (green highlight, page 9, line 271-274)

4. Lines 288-290: “The overall head structures were similar, although there were some regions that deviated, possibly due to the dimerization and allosteric effects induced by 41-6.” There is no evidence to support that 41-6 binding causes head dimerization. This claim should either be further supported or removed.

Response: We appreciate your suggestion and removed “the dimerization” as per your suggestion. (green highlight, page 10, line 293-296)

5. Lines 358-363 are better placed in the Introduction.

Response: We appreciate your suggestion and have transferred this section to the introduction section. (green highlight, page 3, line 64-69)

6. Lines 408-410: recommend rephrasing as follows: “Our structural data revealed two populations where the soluble NiV-RBP ectodomain that has two heads near the viral terminal and two heads near the host terminal, was bound with either one or three 41-6 Fabs.”

Response: We appreciate your suggestion and have made the modification to this description as per your suggestion. (green highlight, page 14, line 404-406)

7. Lines 417-420: “In contrast to non-receptor-competitive antibody nAH1.3, our 41-6, a receptor-competitive antibody, is limited to binding to only one head region, possibly because the binding epitope in the other protomer is obstructed by the adjacent subunit.” This is incorrect. Although the high-resolution structure is of one Fab, the data (Extended Figure 5, panel b) show up to 3 41-6 bound to NiV-RBP.

Response: We are appreciative of your constructive criticism. The purpose of this paragraph is to indicate that 41-6 can only bind to one of the two protomers on the side close to the host cell membrane, as the binding epitope obstructed by neighboring protomer. We have revised the description as follows: “In contrast to non-receptor-competitive antibody nAH1.3, our 41-6, a receptor-competitive antibody, is limited to binding to only one of the two head regions located near the host terminus. This is

possibly because the binding epitope in the other protomer is obstructed by the adjacent subunit.” (green highlight, page 14, line 413-417)

8. Lines 420-423: “This type of binding has not been documented previously, and upon further investigation. We found the epitope of 41-6 is similar to the binding sites in RBP of the receptor and other receptor-like antibodies.” This is a fragmented sentence and should be revised. References must be provided for the other structures that are being compared. A figure must be shown for this comparison.

Response: We are deeply thankful for your suggestion and have made the necessary revision to include reference to the other mentioned structure (Xu et al., 2013; Bowden et al., 2008). This paragraph in the discussion section is intended to summarize the comparison of structure with more detailed information provided in Supplementary Fig. 9d and Supplementary Fig. 10c. (green highlight, page 14, line 417-420)

Reference:

1. Xu, K., Rockx, B., Xie, Y., DeBuysscher, B.L., Fusco, D.L., Zhu, Z., Chan, Y.P., Xu, Y., Luu, T., Cer, R.Z., et al. (2013). Crystal structure of the Hendra virus attachment G glycoprotein bound to a potent cross-reactive neutralizing human monoclonal antibody. PLoS Pathog 9, e1003684.
2. Bowden, T.A., Aricescu, A.R., Gilbert, R.J., Grimes, J.M., Jones, E.Y., and Stuart, D.I. (2008). Structural basis of Nipah and Hendra virus attachment to their cell-surface receptor ephrin-B2. Nat Struct Mol Biol 15, 567-572.

Reviewer #3 (Remarks to the Author):

My prior concerns were addressed in an ok manner. It would have been nice to perform the additional BSL-4 experiments suggested, but I understand the time it will take.

The fonts in several of the figures are still too tiny and hard to read.

Response: We greatly appreciate your efforts in the review process and have tried our best to improve the presentation of the figures.